# Boosting the Transferability of Adversarial Attack on Vision Transformer with Adaptive Token Tuning

**Di Ming, Peng Ren, Yunlong Wang, Xin Feng**[*]
School of Computer Science and Engineering, Chongqing University of Technology
Chongqing, China
diming@cqut.edu.cn, misterr_2019@163.com, {ylwang, xfeng}@cqut.edu.cn

## Abstract

Vision transformers (ViTs) perform exceptionally well in various computer vision tasks but remain vulnerable to adversarial attacks. Recent studies have shown that the transferability of adversarial examples exists for CNNs, and the same holds true for ViTs. However, existing ViT attacks aggressively regularize the largest token gradients to exact zero within each layer of the surrogate model, overlooking the interactions between layers, which limits their transferability in attacking black-box models. Therefore, in this paper, we focus on boosting the transferability of adversarial attacks on ViTs through adaptive token tuning (ATT). Specifically, we propose three optimization strategies: an adaptive gradient re-scaling strategy to reduce the overall variance of token gradients, a self-paced patch out strategy to enhance the diversity of input tokens, and a hybrid token gradient truncation strategy to weaken the effectiveness of attention mechanism. We demonstrate that scaling correction of gradient changes using gradient variance across different layers can produce highly transferable adversarial examples. In addition, introducing attentional truncation can mitigate the overfitting over complex interactions between tokens in deep ViT layers to further improve the transferability. On the other hand, using feature importance as a guidance to discard a subset of perturbation patches in each iteration, along with combining self-paced learning and progressively more sampled attacks, significantly enhances the transferability over attacks that use all perturbation patches. Extensive experiments conducted on ViTs, undefended CNNs, and defended CNNs validate the superiority of our proposed ATT attack method. On average, our approach improves the attack performance by 10.1% compared to state-of-the-art transfer-based attacks. Notably, we achieve the best attack performance with an average of 58.3% on three defended CNNs. Code is available at `https://github.com/MisterRpeng/ATT`.

## 1 Introduction

The Vision Transformer (ViT) [1] was the first to apply the transformer architecture to computer vision models, demonstrating excellent performance in image classification tasks. Since then, various ViT-based transformer structures have shown comparable success in a range of computer vision tasks, including object detection [2, 3], semantic segmentation [4, 5], and human pose estimation [6]. However, similar to convolutional neural networks (CNNs), ViT models are also vulnerable to adversarial attacks [7, 8, 9, 10]. For ViT to be widely adopted in real-world applications, particularly in secure systems, it is crucial to identify model weaknesses [11, 12, 13] and develop more robust ViT models. Specifically, adversarial attacks generate perturbations that can cause incorrect model classifications, which are often too subtle for humans to notice. The transferability of the adversarial

---

[*]Corresponding Author: Xin Feng

38th Conference on Neural Information Processing Systems (NeurIPS 2024).

perturbations leads to the fact that the perturbations can directly deceive the unknown target model. Therefore, it is imperative to study transferable ViT attacks to further uncover the vulnerabilities of ViT models, thereby providing valuable insights for adversarial defense or building robust ViTs.

Gradient regularization-based adversarial attack adjusts the gradient after back-propagation to update the perturbation more stably and effectively. However, attack methods designed specifically for CNNs are less effective when directly applied to ViTs. Token gradient regularization (TGR) [14] addresses the shortcomings of existing gradient regularization methods and designs a technique that regularizes the gradient of intermediate ViT layers, leading to a significant improvement of the transferability of adversarial attacks on ViTs. However, we observe that TGR simply sets the largest part of token gradients in each module to zero, thereby losing feature information to a certain extent. To solve this problem, we propose an adaptive variance reduction method, not only correcting the gradient variance but also preserving the feature information (see the bottom-right of Fig. 1). Thus, the perturbation can learn more features while reducing overfitting and improving the transferability of adversarial examples across ViT models with various network structures.

Another ViT attack approach, proposed by Wei *et al.* [15], inspired us to incorporate input diversity with gradient regularization to enhance the transferability further. This transfer-based ViT method proposes the PatchOut Attack to exploit the input diversity [16]. PatchOut is a straightforward and effective way to address the overfitting. However, this approach does not account for update efficiency and difficulty, leading to potential over-discarding of updates. As a result, the finite number of iterations may end with a perturbation that has not reached its optimal state. To address this, we propose a discard strategy that combines feature importance-based sampling and self-paced learning. As shown on the left side of Fig. 1, this approach allows for a more rational implementation of patch out strategy and improves the update efficiency.

Influenced by [15, 17], we found out that the attention mechanism specific to ViTs tends to cause the perturbation overfitting in surrogate models. To resolve this, we propose an attention weakening strategy to further adaptively tune the token gradients both within and across ViT layers. Specifically, as indicated in the top-right of Fig. 1, we truncate the gradient back-propagation in the attention module of deep ViT layers while preserving the token gradients of QKV and MLP modules, to reduce the overfitting caused by excessive global attention. In addition, for purpose of balancing the interactions between different modules in the remaining shallow ViT layers, we also adjust the token gradients of the attention, QKV and MLP modules accordingly.

In summary, the main contributions of our paper include:
- We propose an adaptive token gradient re-scaling method to improve the transferability of adversarial attacks. This method aims to reduce the overall variance of token gradients in each ViT layer and smooth token gradient variance changes throughout all the ViT layers. Meanwhile, we introduce a hybrid token gradient truncation method to weaken the effectiveness of attention mechanism. In order to achieve a balance between different modules, we further adjust the truncation factors of each module accordingly.
- We propose a self-paced patch out strategy for learning the input diversity. During the perturbation training, a sparse mask is constructed under the semantic guidance of patch-level feature importance. As the iterations increase, self-paced learning strategy is employed to generate a progressively more sampled mask. These sparse masks are then superimposed with perturbations as input samples, and the perturbation updates are incrementally enhanced through this self-paced training process, which not only mitigates the overfitting issue but also enhances the attack efficiency.
- Extensive experiments are conducted to verify the effectiveness of our proposed adaptive token tuning attack method, which improves the attack success rate by an average of 10.1% over state-of-the-art methods under black-box attack setting, including an average of 4.9% for attacking ViTs and 12.8% for attacking CNNs. Additionally, we achieves the highest average attack rate of 58.3% on defended CNNs, further demonstrating the superior transferability of our ATT attack approach in both undefended and defended black-box models.

## 2 Related Work

Current adversarial attacks can be broadly divided into two main categories: white-box and black-box attacks. White-box attack refers to adversarial attack under the condition that all the model information is known. On the contrary, black-box attacks do not have unrestricted access to all

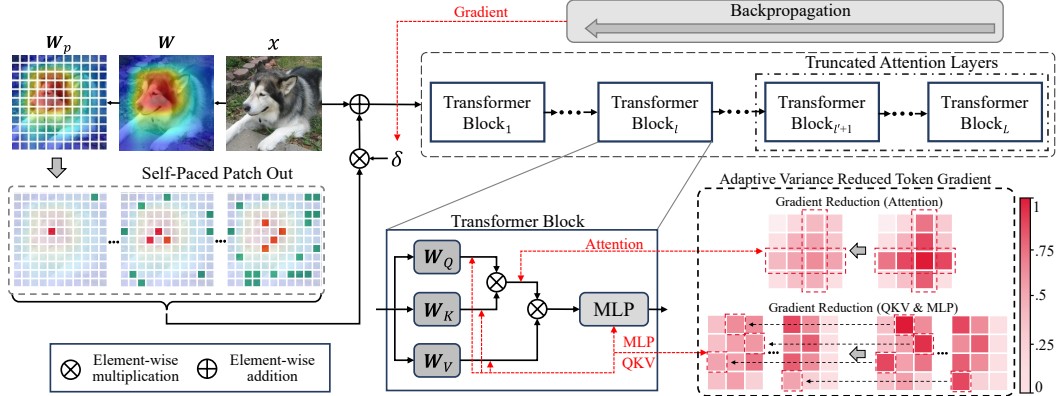

Figure 1: The overall framework of our **A**daptive **T**oken **T**uning (**ATT**) attack method.

model's information. Specifically, black-box attacks consist of two types: query-based black-box attacks, which could access the output of the target model, and transfer-based black-box attacks, where no knowledge of the target model is available. In practice, it is often impossible to obtain any information about the target model, making transfer-based black-box attacks a more serious security threat. Therefore, this paper focuses on transferable black-box attacks on ViT models.

**Adversarial Attacks on CNNs.** Adversarial attacks on CNNs can be categorized into gradient optimization-based and feature information-based attacks. In gradient optimization-based adversarial attacks, [18, 19, 20] incorporated a momentum term into the existing iterative attack update process [21] to stabilize the update direction and avoid the local optima. Towards improving the backpropagation path, Skipping Gradient Method (SGM) [22] allowed gradients to backpropagate more through skip connections, while Backward Propagation Attack (BPA) [23] mitigated the negative impact of gradient truncation in ReLU and max-pooling to improve the relevance between the gradient and the input. Besides, [24, 25] introduced variance adjustment methods that modify the current gradient based on the variance of gradients from the previous iteration and from ensemble surrogate models respectively, resulting in more stable gradient updates. In comparison, feature information-based adversarial attacks, such as FIA [26] and FDA [27], aim to degrade model performance by disrupting important features that play a critical role in the model's decisions. Since inaccurate feature importance guidance could lead to a decrease in transferability, [28, 29] proposed the neuron attribution-based attack methods that utilized more accurate neuron importance for feature-level attacks. Furthermore, there were also other approaches [30, 31, 32] that utilized feature importance to create masks that guide perturbation updates, enhancing transferability by focusing on more critical features during attacks. In addition to that, [33, 34] specifically targeted feature information from shallow layers to craft more transferable perturbation against the fine-tuned models.

**Adversarial Attacks on ViTs.** However, attack methods designed for CNNs have very low transferability to ViT models. To improve the attack performance against ViTs, Nasser *et al.* [35] proposed the self-integration (SE) method and token refinement module (TR). Specifically, SE leverages the classification header at each layer of the ViT model to produce adversarial perturbations, while TR further refines the classification token based on SE to enhance the attack transferability. Wei *et al.* [15] approached the problem from the perspective of attention, proposing a method to skip the attention module, which significantly improves the transferability of adversarial examples. Additionally, they proposed a random extraction of perturbation patch strategy from the perspective of image transformation to further enhance the transferability. Zhang *et al.* [36] introduced a virtual dense connectivity approach, enabling the backpropagation of gradients through jump connections in a deep network. Gao *et al.* [37] developed a feature diversity-based attack that uses adversarial perturbations to accelerate feature collapse due to the ViT attention module. Zhang *et al.* [14] proposed a token gradient regularization approach to reduce gradient variance during backpropagation in a token-wise manner, based on the structural features of ViTs, to improve the transferability of adversarial perturbations. By contrast, our adaptive token tuning method not only considers the token gradient variance but also aims to reduce the destruction of feature diversity to preserve the original feature information. Additionally, we design a self-paced patch out strategy to enhance both the transferability and efficiency of adversarial attacks.

## 3 Methodology

### 3.1 Preliminary

**Notations and Definitions.** A benign sample is defined as the original clean image $x \in \mathbb{R}^{C \times H \times W}$ along with its corresponding ground-truth label $y \in \{1, 2, \cdots, K\}$, where $C$, $H$, $W$, and $K$ represent the number of channels, height, width of the image, and the number of label categories, respectively. In ViT networks, the image $x$ is evenly partitioned into a set of patches $x_p = \{x_p^1, x_p^2, \cdots, x_p^n\}$, where $x_p^i \in \mathbb{R}^{C \times P \times P}$ denotes the $i$-th patch and $n = HW/P^2$ denotes the number of patches. The set of ViT modules is defined as $\mathcal{M} = \{\text{QKV}, \text{MLP}, \text{Attention}\}$, and the total number of ViT layers is $L$. Given a surrogate ViT model $f$, the predicted label for the input $x$ is represented by $\hat{y} = f(x)$, where $\hat{y} = y$ for benign sample $x$. Consistent with previous works, we focus on fooling ViT models in the untargeted attack setting. To guarantee the effectiveness and imperceptibility of adversarial example $x_{adv} = x + \delta$, adversarial attack maximizes the following optimization problem:

$$\arg\max_{\delta} \mathcal{L}(f(x + \delta), y), \quad \text{s.t.} \ \|\delta\|_{\infty} \leq \epsilon \tag{1}$$

where $\mathcal{L}$ is the loss function, *e.g.*, cross-entropy, and $\epsilon$ is a constant to constrain the adversarial perturbation $\delta$ in the $\ell_{\infty}$-norm bound, thereby resulting in a change of label that $f(x_{adv}) \neq y$.

**Patch Out.** Motivated by previous work [16], Wei *et al.* [15] proposed the PatchOut attack to alleviate the overfitting phenomenon by creating diverse input patterns. Specifically, a portion of patches in the adversarial perturbation are randomly discarded using a binarized attack mask $w$, where the values of selected/unselected patches are set to ones/zeros respectively. During the feed-forward process, the input to surrogate model $f$ becomes as $x + \delta \odot w$, thus enhancing the input diversity.

**Token Gradient Regularization.** As compared to gradient variance tuning [24] applied directly to the input, Zhang *et al.* [14] regularized the gradient variance of tokens in the intermediate ViT blocks. Through the back-propagation in ViT layers, token gradients are multiplied by a scaling factor $s$ and $k$ extreme values are further reset to zeros. As a result, adversarial perturbation is updated by a regularized input gradient, *i.e.*, $TGR(\nabla_{\delta}\mathcal{L}, s, k)$, resulting in a more stable optimization process.

### 3.2 Adaptive Variance Reduced Token Gradient

During the training, a large gradient variance tends to overfit the surrogate model and causes the update of the perturbations to fall into a local optimal solution, which lowers the transferability of adversarial attack. Wang *et al.* [24] propose a variance tuning method to reduce the gradient variance in the input perturbation space, but ignoring the gradient variance in intermediate layers. Thus, we aim to reduce the overall gradient variance across all ViT layers to improve training effectiveness.

**Variance Reduction in a Single ViT Layer.** Given the module $m \in \mathcal{M}$ in an intermediate $l$-th ViT layer, the latent representation of $i$-th token is defined as $z_i^{(l,m)}$ and its corresponding gradient *w.r.t.* the loss function $\mathcal{L}$ in Eq. 1 is defined as $g_i^{(l,m)} = \partial\mathcal{L}/\partial z_i^{(l,m)}$. During the back-propagation, TGR [14] searches the largest gradient $g_j^{(l,m)}$ out of $n$ token gradients via $j = \arg\max_{i \in \{1, \cdots, n\}} g_i^{(l,m)}$ for each module $m$, and simply sets $g_j^{(l,m)} = 0$ to reduce the variance of token gradients in $l$-th ViT layer. However, this largest gradient could be highly correlated with important features. As a consequence, iterative optimization updates may overlook this portion of semantic information, undermining the quality of the perturbations.

To simultaneously reduce the token gradient variance and preserve the feature information from the largest token gradient, we decrease the largest token gradient mildly by a gradient penalty factor $\gamma \in (0, 1)$. For $m = \text{QKV}$ or $m = \text{MLP}$, $g_i^{(l,m)} = \gamma \cdot g_i^{(l,m)}$. Following the TGR [14] method, for $m = \text{Attention}$, we also search for the set $\mathcal{S}$ of extreme token gradients located in the same row or column as the largest token gradient. Thus, all the token gradients in $\mathcal{S}$ are re-scaled by $g_i^{(l,m)} = \gamma \cdot g_i^{(l,m)}$ for $i \in \mathcal{S}$, since they are highly correlated with the largest token gradient. As can be seen, this mild re-scaling strategy allows some important feature information in the largest or extreme token gradients (*i.e.*, nonzero gradient values) to back-propagate between consecutive ViT layers in certain degree. The detailed analysis of the change in the token gradient variance (before and after mild re-scaling by $\gamma$) is provided in Appendix A.

**Adaptive Variance Reduction Throughout ViT Layers.** If re-scaling the token gradient of each intermediate ViT layer independently, after back-propagation, the gradient *w.r.t.* the input perturbation could be deviated from the original updating direction. Inspired by [38, 39], we utilize the gradient information across ViT layers to adaptively re-scale the token gradients in each ViT layer. To be specific, the last ViT layer is selected as the anchor, since it is most relevant to the classification task. Then, the largest token gradient (or the set of extreme token gradients) in each ViT layer is adaptively re-scaled to ensure its gradient variance remains consistent with previous ViT layer. Thus, for $t$-th iteration, we define the adaptive variance reduce token gradient method as follows:

$$\boldsymbol{g}_{i,t}^{(l,m)} = \boldsymbol{g}_{i,t}^{(l,m)} \cdot \left(\gamma + \lambda\Big(1 - \sqrt{\boldsymbol{\Phi}_t^{(l,m)}/\boldsymbol{\Phi}_t^{(l+1,m)}}\Big)\right), \tag{2}$$

where $\boldsymbol{\Phi}_t^{(l,m)}$ and $\boldsymbol{\Phi}_t^{(l+1,m)}$ denote the variance of token gradients for module $m$ in $l$-th and $(l+1)$-th layers respectively, *i.e.*, $\boldsymbol{\Phi}_t^{(l,m)} = \mathrm{Var}(\boldsymbol{g}^{(l,m)})$ and $\boldsymbol{\Phi}_t^{(l+1,m)} = \mathrm{Var}(\boldsymbol{g}^{(l+1,m)})$, and $\lambda$ is the adaptive factor balancing the relative importance between the gradient penalty factor and the ratio of gradient variances. With this adaptive updating strategy, the variances of token gradients between consecutive ViT layers become smoother compared to using a fixed constant value for re-scaling token gradients. Appendix B provides further details regarding the analysis of adaptive variance reduced token gradient throughout ViT layers with different parameter settings.

### 3.3 Self-Paced Patch Out under Semantic Guidance

Undoubtedly, random discarding [15] is a simple and effective method to alleviate the overfitting and improve the transferability. However, discarding patches inappropriately raised by randomness could lead to the difficulty in updating the perturbation, especially when limiting the total number of iterations to a small value, *e.g.*, only 10 iterations are commonly used in the literature of ViT attack. Thus, within a limited training budget, we propose a self-paced patch out strategy under semantic guidance to prevent the improper discarding of important patches to a certain degree. In addition to that, the number of discarded patches are dynamically controlled by a scheduled pace to further stabilize the perturbation training.

**Generating Semantic Guided Sparse Mask.** Instead of generating a completely random mask, we leverage the rich semantic information to mitigate the optimization instability caused by discarding perturbation patches. Based on Grad-CAM [40], we construct the feature importance matrix $\boldsymbol{W} \in \mathbb{R}^{H \times W}$ by fusing gradients and features from an intermediate ViT layer, *i.e.*, $\boldsymbol{W} = \sum_{i=1}^{C^{(l)}} \boldsymbol{G}_i^{(l)} \odot \boldsymbol{F}_i^{(l)}$, where $C^{(l)}$ is the number of channels in $l$-th layer and $l \in (0, L)$. According to the partition of $\boldsymbol{x}_p$, we further define the patch version of $\boldsymbol{W}$ as $\boldsymbol{W}_p = \{\boldsymbol{W}_p^1, \cdots, \boldsymbol{W}_p^n\}$, where $\boldsymbol{W}_p^i \in \mathbb{R}^{P \times P}$. Thus, the feature importance of $i$-th patch $\boldsymbol{x}_p^{(i)}$ can be measured by the Frobenius norm $||\boldsymbol{W}_p^i||_F$.

Furthermore, patch-level feature importance can be combined with random sampling to discard a portion of patches under semantic guidance. First, we define $\boldsymbol{1}_p$ as a $C \times P \times P$ tensor with all ones and the scaled patch-level feature importance as $\boldsymbol{c} \in [0,1]^{C \times H \times W}$, where $\boldsymbol{c}_p^i = (||\boldsymbol{W}_p^i||_F - \min_j(||\boldsymbol{W}_p^j||_F))/(\max_j(||\boldsymbol{W}_p^j||_F) - \min_j(||\boldsymbol{W}_p^j||_F)) \cdot \boldsymbol{1}_p$ for $i, j \in \{1, \cdots, n\}$. To control the number of discarded patches, we further introduce $\alpha \geq 0$ and $\beta \geq 0$ as scaling and offset coefficients to shift the distribution of $\boldsymbol{c}$. Then, the semantic guided sparse mask $\boldsymbol{w}$ can be generated by:

$$\boldsymbol{w} = (\boldsymbol{q} < \alpha \cdot \boldsymbol{c} - \beta), \tag{3}$$

where $\boldsymbol{q} \in [0,1]^{C \times H \times W}$ is a random variable sampled from the patch-level uniform distribution $\mathcal{U}_p(0,1)$. In detail, for $i$-th patch of $\boldsymbol{q}$, we have $\boldsymbol{q}_p^i = \epsilon \cdot \boldsymbol{1}_p$ where $\epsilon \sim \mathcal{U}(0,1)$. As can be seen, the elements within a patch will be discarded or preserved altogether with the same probability. Additionally, patches with lower feature importance are more likely to be discarded, while those with higher feature importance are less likely to be discarded.

**Self-Paced Patch Out via Progressive Sparse Mask.** To further improve both the efficiency and the effectiveness of training adversarial perturbations, a self-paced patch out strategy is introduced to control the number of discarded patches for each iteration at a dynamic pace. For $t$-th iteration, we define the progressive sparse mask $\boldsymbol{w}_t$ based on feature importance $\boldsymbol{c}_t$ as follows:

$$\begin{aligned} &\boldsymbol{w}_t = \big\{\boldsymbol{w}|\boldsymbol{w} = (\boldsymbol{q}_t < \alpha \cdot \boldsymbol{c}_t - \beta),\ \boldsymbol{q}_t \sim \mathcal{U}_p(0,1),\ \boldsymbol{c}_t = 1 - \tfrac{t}{T}(1 - \boldsymbol{c})\big\}, \\ &N_p(\boldsymbol{w}_1) < N_p(\boldsymbol{w}_2) < \cdots < N_p(\boldsymbol{w}_T), \end{aligned} \tag{4}$$

where $T$ is the number of iterations, $\boldsymbol{q}_t$ is the random variable sampled from $\mathcal{U}_p(0,1)$, and $N_p(\boldsymbol{w}_t)$ is the number of discarded patches in $\boldsymbol{w}_t$, approximately equal to $\mathbb{E}[1 - \alpha \cdot \boldsymbol{c}_t + \beta)]/(C \cdot P^2)$.

To be specific, as the iteration $t$ increases, $\boldsymbol{c}_t$ becomes to $\boldsymbol{c}$ gradually, and more perturbation patches are discarded such as $N_p(\boldsymbol{w}_t) < N_p(\boldsymbol{w}_{t+1})$ to prevent the overfitting. In the meanwhile, training samples with varying patterns are fed into the surrogate ViT model for purpose of enhancing the diversity of inputs. On the other hand, the coefficients $\alpha$ and $\beta$ can be adjusted correspondingly to guarantee that the strength (*i.e.*, the magnitude) of adversarial perturbations increases step by step, *e.g.*, $||\boldsymbol{w}_t \odot \boldsymbol{\delta}_t||_1 \leq ||\boldsymbol{w}_{t+1} \odot \boldsymbol{\delta}_{t+1}||_1$. For a fair comparison, we also set appropriate values for $\alpha$ and $\beta$ to generate progressive sparse masks, ensuring that the total number of discarded patches throughout the training process is greater than or at least equal to that of other methods such as PNA [15]. Thus, we have $\sum_{t=1}^{T} \mathbb{E}[1 - \alpha \cdot \boldsymbol{c}_t + \beta)]/(C \cdot P^2) \geq T \cdot N_{dpatch}$, where $T$ is the number of iterations and $N_{dpatch}$ is the fixed number of patches discarded by comparison methods per iteration. To summarize, our proposed self-paced patch out effectively integrates self-paced learning into training process and maximizes the adaptive loss function $\mathcal{L}_t = \mathcal{L}(f(\boldsymbol{x} + \boldsymbol{\delta} \odot \boldsymbol{w}_t), y)$ at each iteration, which can further improve the transferability of crafted adversarial perturbations across various models.

### 3.4 Weakening the Effectiveness of Attention Mechanism

Feature-based adversarial attacks [26, 29, 33, 34] have shown that not all the features in DNNs contribute positively to the generation of adversarial perturbations, despite their significances in classification tasks. On the other hand, it is demonstrated in previous ViT studies [15, 17] that attention modules in certain ViT layers are redundant for image classification and adversarial perturbation generation, which could cause the overfitting phenomenon. Building upon this, we propose to reduce the impact of some attention modules during the training of adversarial perturbations.

**Truncated Attention Layers.** According to [41, 35], shallow layers exploit generic properties of attention. whereas deep layers exploit highly model-specific properties of attention. To mitigate the overfitting caused by excessive global attention, we introduce a hard truncation strategy for deep ViT layers. During back-propagation, we multiply the token gradient $g_i^{(l,m)}$ with a truncation factor $\tau$ for module $m = $ Attention. To be specific, we set $\tau^{(l,m)}$ to 0 for $l \in \{l' + 1, \cdots, L\}$, while setting $\tau^{(l,m)}$ to non-zero value for $l \in \{1, \cdots, l'\}$. As a result, intermediate gradients can be back-propagated to each token individually via skipping attention mechanism, thereby reducing the adversarial perturbation's dependence on complex interactions between tokens in deep ViT layers.

**Hybrid Token Gradient Truncation.** For purpose of effectively balancing the influence of different modules on the perturbation training, we further introduce a hybrid token gradient truncation method to constrain the token gradient in each module $m$ throughout all ViT layers using the predefined set $\mathcal{S}_\tau^{(m)}$ of $L$ truncation factors. For $m = $ Attention, we set $\mathcal{S}_\tau^{(m)} = \{\tau^{(1,m)}, \cdots, \tau^{(l',m)}, 0, \cdots, 0\}$, where deep ViT layers are processed via the hard truncation (*i.e.*, $\tau^{(l,m)} = 0, l > l'$), and shallow ViT layers are processed via the soft truncation (*i.e.*, $\tau^{(l,m)} > 0, l \leq l'$). For either $m = $ QKV or $m = $ MLP, we set $\mathcal{S}_\tau^{(m)} = \{\tau^{(1,m)}, \cdots, \tau^{(l',m)}, \tau^{(l'+1,m)}, \cdots, \tau^{(L,m)}\}$, where all the intermediate ViT layers are processed via the soft truncation (*i.e.*, $\tau^{(l,m)} > 0, l \in \{1, \cdots, L\}$). In addition to that, for any $l$-th ViT layer ($1 \leq l \leq l'$), by setting $\tau^{(l,\text{Attention})} < \max(\tau^{(l,\text{QKV})}, \tau^{(l,\text{MLP})})$ between different ViT modules, we can continue to weaken the effectiveness of attention mechanism during the perturbation training process.

In sections 3.2-3.4, we introduced the details of our proposed adaptive token tuning (ATT) attack method from three perspectives. On the one hand, both variance reduction and hybrid truncation aim to adaptively tune the token gradient throughout the back-propagation path. On the other hand, self-paced patch out focuses on adaptively tuning the token diversity in the input perturbation space. The overall optimization algorithm for training adversarial example is provided in Appendix C.

## 4 Experiments

We utilized different surrogate models for comparison, demonstrating versatility and effectiveness of our ATT attack. Ablation experiments were conducted to verify the effectiveness of each component of our approach. Additionally, we analyzed experimental results by examining gradient variance and feature information, providing a statistical explanation for the effectiveness of our method.

### 4.1 Experiment Setup

**Dataset.** We followed the baseline approach [14] by selecting 1,000 random images from different categories in the ILSVRC2012 [42] validation set. All surrogate models can classify the images in our chosen dataset with near-perfect accuracy.

**Models.** We chose four representative ViT models as surrogate models to produce adversarial samples: ViT-B/16 [1], PiT-B [43], CaiTS/24 [44], and Visformer-S [45]. Considering structural differences between ViTs and CNNs, we divided attack scenarios into two categories: attacking ViTs and attacking CNNs. For ViT attack, we used four ViT models as target models to test the transferability of adversarial samples: DeiT-B [46], TNT-S [47], LeViT-256 [48], and ConViT-B [49]. Similarly, we selected four CNN models as target models to verify the transferability of adversarial samples: Inception-v3 (Inc-v3) [50], Inception-v4 (Inc-v4) [51], Inception-ResNet-v2 (IncRes-v2) [51], and ResNet-v2-152 (Res-v2) [52, 53]. To validate the effectiveness against defense models, we further selected three adversarially trained defense models: Inception-v3-ens3 (Inc-v3ens3), Inception-v4-ens4 (Inc-v4ens4), and Inception-ResNet-v2 (IncRes-v2adv) [54, 55].

**Baseline Methods.** Since our approach takes into account the optimization of the gradient, we selected several methods closely related to ours as baselines, including MI-FGSM (MIM) [18], VMI-FGSM (VMI) [24], and SGM [22]. To demonstrate the superiority of our approach over state-of-the-art ViT attacks, we used PNA [15] and TGR [14] for comparison. Our approach also proposes a superior self-paced input diversity method; therefore, we used PatchOut [15] as the baseline.

**Evaluation Metrics.** Consistent with the baseline methodology [14], we used the attack success rate (ASR) as the evaluation metric in all transferability comparison experiments. Additionally, we defined the number of iterations that led to the first-time misclassification by the model as $t$, which we used as a metric for efficiency comparison within a limited training budget $T$. For ASR, higher values ($\uparrow$) indicate better transferability, while for $t$, lower values ($\downarrow$) indicate better efficiency.

**Parameters.** We kept all known parameter settings consistent with [14]. The maximum perturbation amplitude was set to $\epsilon = 16$, and the number of training iterations was set to $T = 10$, resulting in a step size of $\eta = \frac{\epsilon}{T} = 1.6$ for each perturbation update. All comparison methods used momentum as the stabilization update strategy with decay factor $\mu = 1.0$. Hyperparameters specific to each method were kept the same as those set by original methods. The penultimate ViT layer (*i.e.*, $l = L - 1$) is selected to generate patch-level feature importance. We set appropriate values for $\alpha$ and $\beta$ to ensure the expected value of the number of discards in our method was greater than or equal to the number of PatchOut discards, where its optimal $N_{dpatch}$ is 130. We adjusted the images of the whole dataset to $224 \times 224$ and set the patch size to $16 \times 16$. For the adaptive gradient variance reduction strategy, we set the gradient penalty factor to $\gamma = 0.5$ and the adaptive factor to $\lambda = 0.01$. Truncation factor $\tau$ is finetuned with appropriate values to balance different modules for each surrogate model.

### 4.2 Evaluating the Transferability

In this section, we verified the transferability of adversarial perturbations on four ViTs, four undefended CNNs, and three defended CNN models. Specifically, we produced adversarial samples using the four ViT surrogate models, tested the attack success rate on all black-box models, and calculated the average attack success rate across all black-box models (abbreviated as $\text{Avg}_{bb}$).

Firstly, we focus solely on the ViT attack using "Token Gradient-Based Optimization". Here, all comparison methods and "Ours" indicate the pure gradient-based attack "Without Input Diversity Enhancements". In Table 1, we verified the transferability of our method on ViTs. The experimental results indicated that our method achieved nearly 100% attack success rate under white-box settings. In addition, the transferability of our method performed significantly better than all other baseline methods, with an average increase of 6.4% in the attack success rate of black-box models. All baseline methods optimized the gradient and achieved good attack performance, especially TGR, improving the transferability by a large margin. However, TGR directly set the token with the maximum gradient to zero, preventing the perturbation from learning potentially important feature information. In contrast, our method greatly preserved the original feature information while avoiding the overfitting phenomenon caused by excessively large gradients.

Furthermore, we validated the transferability of our method from ViT models to undefended and defended CNN models. Experimental results on CNN models are shown in Table 2, where the

Table 1: The attack success rate (%) of various transfer-based attacks against eight ViT models and the average attack success rate (%) of all black-box models. The best results are highlighted in bold.

| Model | Attack | ViT-B/16 | PiT-B | CaiT-S/24 | Visformer-S | DeiT-B | TNT-S | LeViT-256 | ConViT-B | Avg$_{bb}$ |
|---|---|---|---|---|---|---|---|---|---|---|
| ViT-B/16 | MIM | 100.0* | 34.5 | 64.1 | 36.5 | 64.3 | 50.2 | 33.8 | 66.0 | 49.9 |
| | VMI | 99.6* | 48.8 | 74.4 | 49.5 | 73.0 | 64.8 | 50.3 | 75.9 | 62.4 |
| | SGM | 100.0* | 36.9 | 77.1 | 40.1 | 77.9 | 61.6 | 40.2 | 78.4 | 58.9 |
| | PNA | 100.0* | 45.2 | 78.6 | 47.7 | 78.6 | 62.8 | 47.1 | 79.5 | 62.8 |
| | TGR | 100.0* | 49.5 | 85.0 | 53.8 | 85.6 | 73.1 | 56.5 | 85.4 | 69.8 |
| | Ours | **99.9*** | **57.5** | **90.3** | **63.9** | **90.8** | **82.0** | **66.8** | **90.8** | **77.4↑** |
| PiT-B | MIM | 24.7 | 100.0* | 34.7 | 44.5 | 33.9 | 43.0 | 38.3 | 37.8 | 36.7 |
| | VMI | 38.9 | 99.7* | 51.0 | 56.6 | 50.1 | 57.0 | 52.6 | 51.7 | 51.1 |
| | SGM | 41.8 | 100.0* | 57.3 | 73.9 | 57.9 | 72.6 | 68.1 | 59.9 | 61.6 |
| | PNA | 47.9 | 100.0* | 62.6 | 74.6 | 62.4 | 70.6 | 67.3 | 61.7 | 63.9 |
| | TGR | 60.3 | 100.0* | 80.2 | 87.3 | 78.0 | 87.1 | 81.6 | 76.5 | 78.7 |
| | Ours | **69.6** | 100.0* | **86.1** | **91.9** | **85.5** | **93.5** | **89.0** | **85.5** | **85.9↑** |
| CaiT-S/24 | MIM | 70.9 | 54.8 | 99.8* | 55.1 | 90.2 | 76.4 | 54.8 | 88.5 | 70.1 |
| | VMI | 76.3 | 63.6 | 98.8* | 67.3 | 88.5 | 82.3 | 67.0 | 88.1 | 76.2 |
| | SGM | 86.0 | 55.8 | 100.0* | 68.2 | 97.7 | 91.1 | 74.9 | 96.7 | 81.5 |
| | PNA | 82.4 | 60.7 | 99.7* | 67.7 | 95.7 | 86.9 | 67.1 | 94.0 | 79.2 |
| | TGR | 88.2 | 66.1 | 100.0* | 75.4 | 98.8 | 92.8 | 74.7 | 97.9 | 84.8 |
| | Ours | **93.6** | **76.4** | 100.0* | **85.9** | **99.4** | **96.9** | **87.4** | **98.8** | **91.2↑** |
| Visformer-S | MIM | 28.1 | 50.4 | 41.0 | 99.9* | 36.9 | 51.9 | 49.4 | 39.6 | 42.5 |
| | VMI | 39.2 | 60.0 | 56.6 | 100.0* | 54.1 | 62.8 | 59.1 | 54.4 | 55.2 |
| | SGM | 18.8 | 41.8 | 34.9 | 100.0* | 31.2 | 52.1 | 52.7 | 29.5 | 37.3 |
| | PNA | 35.4 | 61.5 | 54.7 | 100.0* | 51.0 | 66.3 | 64.5 | 50.7 | 54.9 |
| | TGR | 41.2 | 70.3 | 62.0 | 100.0* | 59.5 | 74.7 | 74.8 | 56.2 | 62.7 |
| | Ours | **44.7** | **70.9** | **68.7** | 100.0* | **66.4** | **78.8** | **80.9** | **58.4** | **67.0↑** |

Table 2: The attack success rate (%) of various transfer-based attacks against four undefended CNN models and three defended CNN models and the average attack success rate (%) of all black-box models. The best results are highlighted in bold.

| Model | Attack | Inc-v3 | Inc-v4 | IncRes-v2 | Res-v2 | Inc-v3ens3 | Inc-v3ens4 | IncRes-v2adv | Avg$_{bb}$ |
|---|---|---|---|---|---|---|---|---|---|
| ViT-B/16 | MIM | 31.7 | 28.6 | 26.1 | 29.4 | 22.3 | 19.8 | 16.5 | 24.9 |
| | VMI | 43.1 | 41.6 | 37.9 | 42.6 | 31.4 | 30.6 | 25.0 | 36.0 |
| | SGM | 31.5 | 27.7 | 23.8 | 28.2 | 20.8 | 18.0 | 14.3 | 23.5 |
| | PNA | 42.7 | 37.5 | 35.3 | 39.5 | 29.0 | 27.3 | 22.6 | 33.4 |
| | TGR | 47.5 | 42.3 | 37.6 | 43.3 | 31.5 | 30.8 | 25.6 | 36.9 |
| | Ours | **53.3** | **49.0** | **45.4** | **51.5** | **38.1** | **36.7** | **33.1** | **43.9↑** |
| PiT-B | MIM | 36.3 | 34.8 | 27.4 | 29.6 | 19.0 | 18.3 | 14.1 | 25.6 |
| | VMI | 47.3 | 45.4 | 40.7 | 43.4 | 35.9 | 34.4 | 29.7 | 39.5 |
| | SGM | 50.6 | 45.4 | 38.4 | 41.9 | 25.6 | 20.8 | 16.7 | 34.2 |
| | PNA | 59.3 | 56.3 | 49.8 | 53.0 | 33.3 | 32.0 | 25.5 | 44.2 |
| | TGR | 72.1 | 69.8 | 65.1 | 64.8 | 43.6 | 41.5 | 32.8 | 55.7 |
| | Ours | **80.4** | **75.3** | **72.7** | **72.9** | **52.5** | **50.6** | **41.0** | **63.6↑** |
| CaiT-S/24 | MIM | 48.4 | 42.9 | 39.5 | 43.8 | 30.8 | 27.6 | 23.3 | 36.6 |
| | VMI | 58.5 | 50.9 | 48.2 | 52.0 | 38.1 | 36.1 | 30.1 | 44.8 |
| | SGM | 53.5 | 45.9 | 40.2 | 45.9 | 30.8 | 28.5 | 21.0 | 38.0 |
| | PNA | 57.2 | 51.8 | 47.7 | 51.6 | 38.4 | 36.2 | 30.1 | 44.7 |
| | TGR | 60.3 | 52.9 | 49.3 | 53.4 | 39.6 | 37.0 | 31.8 | 46.3 |
| | Ours | **73.9** | **66.0** | **66.3** | **66.4** | **54.6** | **52.1** | **43.9** | **60.5↑** |
| Visformer-S | MIM | 44.5 | 42.5 | 36.6 | 39.6 | 24.4 | 20.5 | 16.6 | 32.1 |
| | VMI | 54.6 | 53.2 | 48.5 | 52.2 | 33.0 | 32.0 | 22.2 | 42.2 |
| | SGM | 43.2 | 41.1 | 29.6 | 35.7 | 16.1 | 13.0 | 8.2 | 26.7 |
| | PNA | 55.9 | 54.6 | 46.0 | 51.7 | 29.3 | 26.2 | 21.1 | 40.7 |
| | TGR | 65.9 | 66.8 | 55.3 | 60.9 | 36.0 | 32.5 | 23.3 | 48.7 |
| | Ours | **80.9** | **81.2** | **70.5** | **75.7** | **50.1** | **41.3** | **32.0** | **61.7↑** |

transferability of all other methods on CNNs was low, but our method significantly outperformed all baselines. The black-box attack success rate increased by an average of 10.5%, which was higher than the transferability improvement on the ViT model, indicating that our method is more advantageous for cross-model attacks. We analyzed that our method, in addition to avoiding overfitting surrogate models, allowed the perturbation to learn as many features as possible, thus enabling the perturbation to better attack both ViTs and CNN models. Our method achieved the average attack success rate of three surrogate models on CNNs to exceed 60%, with the highest average attack success rate on the defended CNNs reaching 50.2%, which poses a serious security threat. Additional attack results against robust ViTs [56, 57, 58] are provided in Appendix D.9

Lastly, we further investigated the ViT attack strategy that combined "Token Gradient-based Optimization" with "Input Diversity Enhancement", where Patch Out (PO) and the proposed Self-Paced

Table 3: The average attack success rate (%) against ViTs, CNNs, and defended CNNs by various transfer-based attacks with input diversity enhancement strategy. The best results are highlighted in bold. "PO" denotes PatchOut and "SPPO" denotes self-paced patch out under semantic guidance.

| Model | Attack | ViTs | CNNs | Def-CNNs | Model | Attack | ViTs | CNNs | Def-CNNs |
|---|---|---|---|---|---|---|---|---|---|
| ViT-B/16 | MIM+PO | 61.3 | 31.3 | 21.7 | CaiT-S/24 | MIM+PO | 70.3 | 44.0 | 29.3 |
| | VMI+PO | 69.1 | 42.8 | 30.9 | | VMI+PO | 76.8 | 57.8 | 38.4 |
| | SGM+PO | 64.8 | 29.2 | 18.9 | | SGM+PO | 85.1 | 49.2 | 29.3 |
| | PNA+PO | 70.8 | 42.6 | 29.9 | | PNA+PO | 81.6 | 56.6 | 39.3 |
| | TGR+PO | 76.0 | 46.7 | 33.3 | | TGR+PO | 88.8 | 60.5 | 40.5 |
| | Ours+PO | 77.1 | 51.7 | 37.1 | | Ours+PO | 91.1 | 71.9 | 54.3 |
| | Ours+SPPO | 80.3↑ | 54.1↑ | 38.7↑ | | Ours+SPPO | 92.6↑ | 75.4↑ | 58.3↑ |

| Model | Attack | ViTs | CNNs | Def-CNNs | Model | Attack | ViTs | CNNs | Def-CNNs |
|---|---|---|---|---|---|---|---|---|---|
| PiT-B | MIM+PO | 47.3 | 32.5 | 17.5 | Visformer-S | MIM+PO | 54.9 | 45.7 | 23.4 |
| | VMI+PO | 59.5 | 46.2 | 35.8 | | VMI+PO | 64.8 | 56.6 | 32.6 |
| | SGM+PO | 70.0 | 45.6 | 21.3 | | SGM+PO | 51.6 | 44.3 | 15.0 |
| | PNA+PO | 73.1 | 57.8 | 32.7 | | PNA+PO | 68.8 | 61.8 | 32.3 |
| | TGR+PO | 82.3 | 68.9 | 41.3 | | TGR+PO | 70.4 | 64.3 | 33.5 |
| | Ours+PO | 84.2 | 75.2 | 48.4 | | Ours+PO | 70.5 | 79.3 | 44.5 |
| | Ours+SPPO | 87.7↑ | 78.0↑ | 52.0↑ | | Ours+SPPO | 76.4↑ | 84.4↑ | 50.3↑ |

Patch Out (SPPO) were used as the input diversity enhancement strategy. As a result, we denoted these methods with "+PO" or "+SPPO" to indicate that gradient-based attacks were combined "With PO-based or SPPO-based Input Diversity Enhancement". As shown in Table 3, the experimental results indicated that our method demonstrated an average improvement of 6.6% in the black-box average attack success rate against ViTs, CNNs, and defended CNNs (Def-CNNs). When combined with the proposed self-paced patch out, our method exhibited superior attack abilities. Our strategy, which discards perturbation patches under semantic guidance in a self-paced way, proved to be more effective than PatchOut's random discarding. The success rate of our method's black-box attacks was further enhanced by 3.5% when compared to direct utilization of the PatchOut strategy.

In addition to that, we further conducted cross-task transferability experiments on object detections [59, 60, 61, 62, 63] and semantic segmentations [64, 65, 66] in black-box attack scenarios. Adversarial samples crafted by our approach demonstrated greater aggressiveness than baselines, see Appendix D.6 and D.7.

## 4.3 Ablation Study

In this section, we performed three sets of ablation studies on the proposed ATT attack. In the subsequent experiments, all methods were compared based on their complete attack strategies.

**Adaptive Factor.** We explored the role of the adaptive factor $\lambda$ in smoothing the gradient variance across different layers of ViT-B. The attention's gradient is used as a demonstration that similar results can be obtained for other modules. We found that the larger the $\lambda$, the more effective the gradient variance smoothing was. However, the results of the transferability experiments showed that smoother gradient variance did not always result in higher transferability. We analyzed that this might be due to excessive gradient variance smoothing leading to changes and loss of feature information. Therefore, to balance performance, we chose $\lambda = 0.01$. More results are given in Appendix B.

**Attentional Truncation.** Considering the differences between models, we conducted experiments on varying the number of truncated ViT layers. In Fig. 5 of Appendix D.2, three models required truncation to achieve better transferability. Visformer-S achieved the best result without truncation, since it is a shallow model, where scaling alone was sufficient to mitigate overfitting in attention mechanism.

Table 4: The average attack success rate (%) against ViTs, CNNs, and defended CNNs by our method with different module settings.

| Attention | QKV | MLP | ViTs | CNNs | Def-CNNs |
|---|---|---|---|---|---|
| - | - | - | 49.9 | 29.1 | 19.3 |
| ✓ | - | - | 70.4 | 43.3 | 29.7 |
| - | ✓ | | 68.8 | 40.7 | 27.7 |
| - | - | ✓ | 67.8 | 39.5 | 28.0 |
| ✓ | ✓ | - | 72.1 | 45.5 | 31.0 |
| ✓ | - | ✓ | 79.3 | 51.7 | 37.5 |
| - | ✓ | ✓ | 77.8 | 48.7 | 35.5 |
| ✓ | ✓ | ✓ | 80.3 | 54.1 | 38.7 |

**Attacks with Different Module Settings.** We conducted ablation experiments using four surrogate models to compute the average attack success rate for ViTs, undefended CNNs, and defended CNNs in black-box setting. Table 4 shows experimental results on the surrogate model ViT-B, where attacking the attention module greatly improved the transferability, and attacking all three modules achieves the best ASR. The rest of the experimental results are shown in the Appendices D.5 and D.8.

## 4.4    The Analysis of Attack's Effectiveness and Efficiency

We utilized ViT-B/16 to generate adversarial examples and retained intermediate results for 10 iterations. Fig. 2 shows the process of discarding perturbation patches for "Ours+SPPO" versus "TGR+PO", and the classification results (label and probability). For each iteration, class activation map of adversarial example was generated by GradCAM on CaiT-S/24 with an unlabeled setup. The results showed that our method required only 5 iterations for the model to determine the error, while TGR required 9 iterations. Class activation maps also reflected this change. After 10 iterations, the confidence level of wrong label due to our attack was higher, indicating that our attack was not only more efficient but also stronger. Additionally, considering that each iteration increases a fixed perturbation magnitude, our method required a smaller perturbation to mislead the model. More efficiency results tested in different surrogate models are provided in Appendix D.1 and D.3.

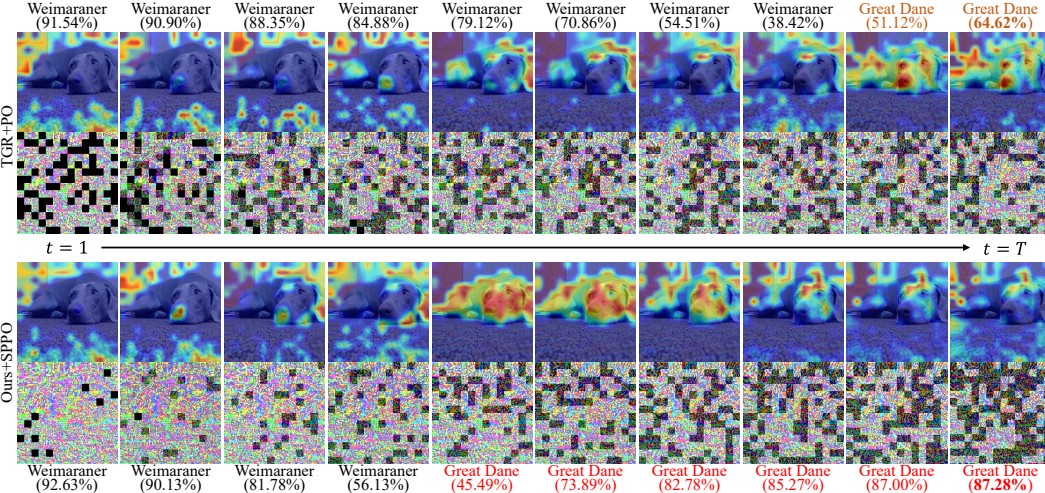

Figure 2: The attack efficiency of "Ours+SPPO" versus "TGR+PO". Our ATT attack makes the model predict wrong faster and ends up generating error labels with a higher level of confidence.

## 5    Conclusion

In this paper, we propose an adaptive token tuning attack method to enhance the transferability of ViT attacks. Unlike previous gradient-based attacks, our method puts more emphasis on smoothing the token gradient variance between different layers and preserving important features along with the back-propagation. Guided by patch-level feature importance, we introduce a self-paced discarding strategy from the perspective of input diversity, where the number of discarded perturbation patches is gradually increased. To further improve transferability, we propose a hybrid truncation strategy to reduce overfitting in the attention mechanism. Extensive experiments show that our adaptive token tuning attack method has superior transferability and efficiency.

## 6    Limitations and Broader Impacts

Although our experimental results verified the effectiveness of the proposed adaptive token tuning strategy in enhancing the transferability of ViT attacks, the relationship between gradient variance reduction and transferability still lacked theoretical support. Existing research suggested that more generalized feature information could improve perturbation's transferability, and our work focused on leveraging this by reducing token gradient variance. In future work, we will continue exploring from a theoretical perspective to provide valuable insights into adversarial attacks.

If the proposed ATT attack method is maliciously used in real-world applications, it could lead to security concerns, representing one of its potential negative social impacts. Due to the superior performance of pretrained models and the significant time cost associated with training from scratch, many applications opt to fine-tune pretrained models. As a result, this undoubtedly exposes applications to a risky environment vulnerable to attacks. Our research aims to encourage deep learning practitioners to further explore security concerns related to model vulnerabilities, and in return, offering constructive guidance for adversarial defense.

## Acknowledgements

This work is partially supported by the National Natural Science Foundation of China (Grant No. 62441607), the Natural Science Foundation of Chongqing, China (Grant No. CSTB2024NSCQ-MSX0341), the Science and Technology Research Program of Chongqing Municipal Education Commission (Grant No. KJQN202301142), the Scientific Research Foundation of Chongqing University of Technology (Grant No. 2022ZDZ026), the Chongqing Education Ministry's Key Research Base Project for Humanities and Social Sciences (Grant No. 24SKJD134).

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

# Appendix

## A    The Analysis of the Variance of Token Gradients in ViT Layers

Based on the TGR [14] method's assumption that channels in the ViT module are independent, we can analyze the variance of token gradient in a single channel, which can then be applied repeatedly to all remaining channels within each module. In the following derivations, we replace $\boldsymbol{g}^{(l,m)}[c]$ in $c$-th channel ($c = 1, \cdots, C$) with $\boldsymbol{g}$ for notational simplicity. Without loss of generality, the number of token gradients in each module $m$ is represented by $n$. Thus, $\boldsymbol{g}$ is defined in the $\mathbb{R}^n$ space. Since the scales of different token gradients vary significantly, we can assume that the largest token gradient (either before or after re-scaling) is far beyond the expectation of token gradients, *i.e.*, $\max(\boldsymbol{g}) > 0$, $\max(\boldsymbol{g}) \gg \mathbb{E}[\boldsymbol{g}]$, and $\gamma \cdot \max(\boldsymbol{g}) \gg \mathbb{E}[\boldsymbol{g}]$ with $\gamma \in (0, 1)$.

**Theorem 1.** *Given the token gradients $\boldsymbol{g} = [g_1, \cdots, g_n]$ and the scaling factor $\gamma \in (0,1)$, the variance of token gradients Var$(\boldsymbol{g})$ is reduced after re-scaling the largest token gradient by $g_j = \gamma \cdot g_j$, where $j = \arg\max_{i \in \{1, \cdots, n\}} g_i$.*

**Proof of Theorem 1.** Based on the predefined re-scaling operation on the largest token gradient, we define an auxiliary variable $\widetilde{\boldsymbol{g}} = [\widetilde{g}_1, \cdots, \widetilde{g}_n]$, where $\widetilde{g}_i = \gamma \cdot g_i$ for $i = j$, and $\widetilde{g}_i = g_i$ for $i \neq j$. After re-scaling the largest token gradient by $\widetilde{g}_j = \gamma \cdot g_j$, we can rewrite the variance of the auxiliary gradient variable $\widetilde{\boldsymbol{g}}$ equivalently as follows:

$$
\begin{aligned}
\text{Var}(\widetilde{\boldsymbol{g}}) &= \mathbb{E}[\widetilde{\boldsymbol{g}}^2] - (\mathbb{E}[\widetilde{\boldsymbol{g}}])^2 \\
&= \frac{1}{n}\Big(\sum_{\substack{i=1 \\ i \neq j}}^{n} g_i^2 + \gamma^2 \cdot g_j^2\Big) - \Big[\frac{1}{n}\big(\sum_{\substack{i=1 \\ i \neq j}}^{n} g_j + \gamma \cdot g_j\big)\Big]^2 \\
&= \frac{1}{n}\Big[\sum_{i=1}^{n} g_i^2 - (1-\gamma^2)\cdot g_j^2\Big] - \Big(\frac{1}{n}\sum_{i=1}^{n} g_j - \frac{1-\gamma}{n}\cdot g_j\Big)^2 \\
&= \mathbb{E}[\boldsymbol{g}^2] - (\mathbb{E}[\boldsymbol{g}])^2 - \underbrace{\Big(\frac{1-\gamma}{n}\cdot g_j\Big)\Big[(1+\gamma)\cdot g_j + \frac{1-\gamma}{n}\cdot g_j - 2\mathbb{E}[\boldsymbol{g}]\Big]}_{\Delta}.
\end{aligned} \tag{5}
$$

On the other hand, we have the following inequalities that $g_j \gg \mathbb{E}[\boldsymbol{g}]$ and $\gamma \cdot g_j \geq \mathbb{E}[\boldsymbol{g}]$ since $g_j$ is an outlier (*i.e.*, largest) value from the original distribution of token gradients. As a result, we can obtain that $\widetilde{g}_j \geq \mathbb{E}[\widetilde{\boldsymbol{g}}]$, since $\widetilde{g}_j = \gamma \cdot g_j \geq \mathbb{E}[\boldsymbol{g}] \geq \mathbb{E}[\widetilde{\boldsymbol{g}}]$.

Then, the third term $\Delta$ in the last row of Eq. 5 can be rewritten equivalently as:

$$
\begin{aligned}
\Delta &= \Big(\frac{1-\gamma}{n}\cdot g_j\Big)\Big[(g_j - \mathbb{E}[\boldsymbol{g}]) + \Big(\gamma \cdot g_j + \frac{(1-\gamma)}{n}\cdot g_j - \mathbb{E}[\boldsymbol{g}]\Big)\Big] \\
&= \Big(\frac{1-\gamma}{n}\cdot g_j\Big)\Big[(g_j - \mathbb{E}[\boldsymbol{g}]) + (\widetilde{g}_j - \mathbb{E}[\widetilde{\boldsymbol{g}}])\Big] \geq 0,
\end{aligned} \tag{6}
$$

where $g_j \geq 0$ and $1 - \gamma \geq 0$.

Combining Eq. 5 and Eq. 6 together, we can obtain the following inequality between the original variance and the variance after re-scaling token gradients:

$$
\text{Var}(\widetilde{\boldsymbol{g}}) = \text{Var}(\boldsymbol{g}) - \Delta \leq \text{Var}(\boldsymbol{g}), \tag{7}
$$

which completes the proof.  $\square$

For QKV and MLP modules, we can re-scale the largest token gradient in each channel to reduce the overall gradient variance according to Theorem 1. For Attention module, we found out that the largest token gradient is highly correlated with other extreme token gradients. We define $\mathcal{S}$ as the set of extreme token gradients and its size is $|\mathcal{S}| = n' \ll n$, where $\mathcal{S}$ includes the largest token gradient $g_j$ and its correlated extreme token gradients at the same row or the same column. Thus, we can also assume that all the token gradients in $\mathcal{S}$ are far beyond the expectation of token gradients, *i.e.*, $g_j > 0$, $g_j \gg \mathbb{E}[\boldsymbol{g}]$, and $\gamma \cdot g_j \geq \mathbb{E}[\boldsymbol{g}]$ for $j \in \mathcal{S}$.

**Lemma 2.** *Given the token gradients $\boldsymbol{g} = [g_1, \cdots, g_n]$ and the scaling factor $\gamma \in (0, 1)$, the variance of token gradients $Var(\boldsymbol{g})$ is reduced after re-scaling all the extreme token gradients in the set $\mathcal{S}$ by $g_j = \gamma \cdot g_j$, where $j \in \mathcal{S}$ and $|\mathcal{S}| = n' \ll n$.*

**Proof of Lemma 2.** In the following, the token gradients in the set $\mathcal{S}$ are re-scaled one by one *w.r.t.* their descending order $\mathcal{O}$, *i.e.*, $g_{\mathcal{O}_1} \geq \cdots \geq g_{\mathcal{O}_{n'}}$. For each index $\mathcal{O}_j$, we define an auxiliary gradient variable $\widetilde{\boldsymbol{g}}^{(j)}$ and its corresponding index set $\mathcal{S}^{(j)} = \{\mathcal{O}_1, \cdots, \mathcal{O}_j\}$, where $\widetilde{g}_i^{(j)} = \gamma \cdot g_i$ for $i \in \mathcal{S}^{(j)}$ and $\widetilde{g}_i^{(j)} = g_i$ for $i \notin \mathcal{S}^{(j)}$. For any two consecutive indices $\mathcal{O}_j$ and $\mathcal{O}_{j+1}$, the relationships between gradients and their expectations are $\widetilde{g}_{\mathcal{O}_j} \geq \mathbb{E}[\widetilde{\boldsymbol{g}}^{(j)}]$ and $\widetilde{g}_{\mathcal{O}_{j+1}} \geq \mathbb{E}[\widetilde{\boldsymbol{g}}^{(j+1)}]$.

According to Theorem 1, we have the following inequality of variances before and after re-scaling $\widetilde{g}_{\mathcal{O}_{j+1}}^{(j+1)} = \gamma \cdot \widetilde{g}_{\mathcal{O}_{j+1}}^{(j)}$, *i.e.*, $Var(\widetilde{\boldsymbol{g}}^{(j+1)}) \leq Var(\widetilde{\boldsymbol{g}}^{(j)})$. By setting $\widetilde{\boldsymbol{g}}^{(0)} = \boldsymbol{g}$ and iteratively re-scaling token gradients in the set $\mathcal{S}$ using the descending order $\mathcal{O}$, the following inequality can be obtained:

$$Var(\boldsymbol{g}) = Var(\widetilde{\boldsymbol{g}}^{(0)}) \geq Var(\widetilde{\boldsymbol{g}}^{(1)}) \geq \cdots \geq Var(\widetilde{\boldsymbol{g}}^{(n'-1)}) \geq Var(\widetilde{\boldsymbol{g}}^{(n')}) = Var(\widetilde{\boldsymbol{g}}), \qquad (8)$$

where $\widetilde{g}_i = \gamma \cdot g_i$ for $i \in \mathcal{S}$ and $\widetilde{g}_i = g_i$ for $i \notin \mathcal{S}$. As a result, we have $Var(\widetilde{\boldsymbol{g}}) \leq Var(\boldsymbol{g})$ after re-scaling all the $n'$ extreme token gradients in the set $\mathcal{S}$, which completes the proof. $\square$

Based on Lemma 2, we can re-scale the set $\mathcal{S}$ of extreme token gradients in each channel to reduce the overall gradient variance for Attention module. As a result, by combining Theorem 1 and Lemma 2, we theoretically verify the effectiveness of our mild re-scaling strategy that the overall gradient variance can be reduced for all the modules in ViT layers.

## B    The Analysis of Adaptive Variance Reduced Token Gradient

It is demonstrated in Fig. 3a that the variance of token gradients after re-scaling was reduced greatly compared to the variance of original token gradients. Additionally, by adaptively re-scaling the token gradient, the variances throughout ViT layers became more smoother with the increase of the value of $\lambda$, as shown in Fig. 3b. Thus, the proposed adaptive variance reduced token gradient method was more effective than the one of re-scaling the token gradient with a fixed constant value.

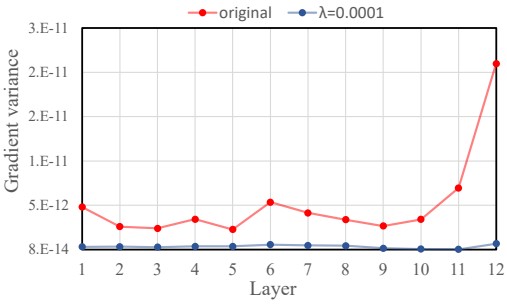
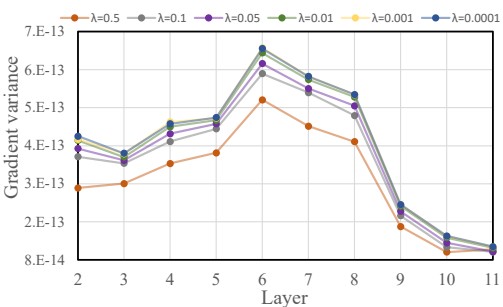

(a) Comparison of original and re-scaled gradients  (b) Comparison of different values of $\lambda$.

Figure 3: The analysis of adaptive variance reduced token gradient with different parameter settings.

## C    The Overall Framework of Optimization Algorithm

In section 3, we present the details of the proposed adaptive token tuning (ATT) method from three perspectives, towards training transferable adversarial examples on ViT models.

At each iteration, during the feed-forward process, self-paced patch out method is applied to randomly discard a portion of adversarial perturbations via the semantic guided sparse mask. Following this, the adaptive loss function and its gradient are calculated based on the input pair (*i.e.*, the crafted adversarial example and the ground truth label). On the other hand, during the back-propagation process, we first truncate token gradient in a hybrid mode to weaken the effectiveness of attention mechanism, and then adaptively re-scale the set of extreme token gradients to reduce the overall gradient variance for each module throughout ViT layers.

The overall optimization algorithm is summarized in Algorithm 1. In line 11, we adaptively re-scale the largest token gradient (*i.e.*, $|\mathcal{S}| = 1$) for MLP or QKV module and the set $\mathcal{S}$ of extreme token gradients for Attention module respectively.

---

**Algorithm 1** Adversarial Attack on ViTs with Adaptive Token Tuning (ATT)

---

**Input:** clean image $x$ with its ground truth label $y$, surrogate model $f$, loss function $\mathcal{L}$, maximum perturbation bound $\epsilon$, maximum iteration number $T$, total number of network layers $L$, gradient penalty factor $\gamma$, adaptive factor $\lambda$, scaling and offset factors $\alpha$, $\beta$, truncation factor $\tau$

**Output:** $x_{adv}$

 1: Initialize $\delta_0 = 0$
 2: Set $\eta = \frac{\epsilon}{T}$
 3: Compute the patch-level feature importance $c$
 4: **for** $t = 1$ to $T$ **do**
 5:     Generate the progressive sparse mask $w_t$ via Eq. 4
 6:     Compute the loss function $\mathcal{L}_t = \mathcal{L}(f(x + \delta_{t-1} \odot w_t), y)$
 7:     Back-propagate the gradient $\nabla \mathcal{L}_t$
 8:     **for** $l = L$ to 1 **do**
 9:         **for** $m \in \{\text{Attention}, \text{MLP}, \text{QKV}\}$ **do**
10:            Truncate the token gradients $g^{(l,m)}$ by $\tau^{(l,m)}$
11:            Adaptively re-scale the set $\mathcal{S}$ of extreme token gradients via Eq. 2
12:         **end for**
13:     **end for**
14:     Update adversarial perturbation $\delta_t = \delta_{t-1} + \eta \cdot \text{sgn}(\nabla_\delta \mathcal{L}_t)$
15: **end for**
16: Generate adversarial example $x_{adv} = \max(\min(x + \delta_T, 255), 0)$
17: **return** $x_{adv}$

---

# D   Additional Experiments

## D.1   Comparison of Attack Efficiency between Different Methods

Tables 5-6 demonstrated the attack efficiency of different methods on ViTs, CNNs, and defended CNNs. To effectively evaluate the attack efficiency during the perturbation training, we calculated the average of the number of iterations that lead to the first-time misclassification by the model across the entire dataset, defined as $t_{avg} = (1/|D|) \cdot \sum_{i=1}^{|D|} t_i$ where $|D|$ represents the total number of test data samples in the dataset $D$. Compared to state-of-the-art methods, experimental results showed that the attack efficiency of our ATT method is the highest in all black-box settings. In Fig. 4, we showed the attack efficiency of different methods in another way, where the subtitle represented the surrogate model and the $x$-axis represented the target model (total 15 models; $0 \sim 7$ are ViTs; $8 \sim 14$ are CNNs). It can be clearly seen that all methods were significantly less efficient against CNNs, which was consistent with the above-mentioned experimental results presented by ASR (%).

## D.2   The Analysis of Truncating Attention Module

Inspired by [17], we considered the possible redundancy of the attention module, which could lead to perturbation overfitting in the surrogate ViT model. Thus, we adopted a truncation strategy to discard part of the attention layer. The corresponding number of truncation layers is chosen based on the best ASR, as shown in Fig. 5, where we conducted the analysis on four surrogate ViT models. Due to differences in network structures, the shallow models (*e.g.*, Visformer-S) obtained the best performance without applying truncation, while the deep models (*e.g.*, PiT-B, CaiT-S/24, Visformer-S) required truncation at varying levels.

Without loss of generality, we set all the non-zero truncation factors of each module to the same value. Additionally, to ensure the effectiveness of hybrid truncation, we first normalized the token gradients of Attention, QKV, and MLP modules to approximately same order of magnitude, *i.e.*, $||g^{(l,m)}||_2/(CHW)$. Then, according to the normalized values, we adjusted the truncation factors to let MLP and QKV modules dominate the back-propagated token gradients and weaken the impact of Attention module on updating the perturbation.

Table 5: Comparison of attack efficiency of different methods on ViTs.

| Model | Attack | ViT-B/16 | PiT-B | CaiT-S/24 | Visformer-S | DeiT-B | TNT-S | LeViT-256 | ConViT-B |
|---|---|---|---|---|---|---|---|---|---|
| ViT-B/16 | PNA+PO | 2.3 | 7.8 | 5.6 | 7.5 | 5.8 | 6.1 | 7.8 | 5.6 |
| | TGR+PO | 2.6 | 7.5 | 5.8 | 7.2 | 5.8 | 5.4 | 7.3 | 5.6 |
| | Ours+SPPO | **1.3**↓ | **6.3**↓ | **4.1**↓ | **6.0**↓ | **4.1**↓ | **4.3**↓ | **6.1**↓ | **3.9**↓ |
| PiT-B | PNA+PO | 7.5 | 1.3 | 6.9 | 5.7 | 7.1 | 5.5 | 6.4 | 6.8 |
| | TGR+PO | 5.9 | 1.8 | 5.7 | 4.9 | 5.9 | 4.1 | 5.1 | 5.7 |
| | Ours+SPPO | **5.3**↓ | **0.9**↓ | **4.5**↓ | **3.7**↓ | **4.7**↓ | **3.2**↓ | **4.0**↓ | **4.6**↓ |
| CaiT-S/24 | PNA+PO | 5.9 | 7.0 | 2.3 | 6.5 | 4.4 | 5.1 | 6.6 | 4.5 |
| | TGR+PO | 4.7 | 6.3 | 1.9 | 5.7 | 3.8 | 3.9 | 5.5 | 3.8 |
| | Ours+SPPO | **3.5**↓ | **4.9**↓ | **0.9**↓ | **4.3**↓ | **2.4**↓ | **2.7**↓ | **4.2**↓ | **2.4**↓ |
| Visformer-S | PNA+PO | 8.0 | 6.5 | 7.3 | 0.8 | 7.6 | 5.5 | 6.2 | 7.6 |
| | TGR+PO | 6.8 | 5.8 | 6.3 | 1.1 | 6.6 | 5.1 | 6.7 | 6.9 |
| | Ours+SPPO | **6.5**↓ | **5.4**↓ | **5.8**↓ | **0.8**↓ | **6.2**↓ | **3.8**↓ | **4.6**↓ | **6.4**↓ |

Table 6: Comparison of attack efficiency of different methods on CNNs and defended CNNs.

| Model | Attack | IncV3 | IncV4 | IncRes-v2 | ResV2 | IncV3ens3 | IncV3ens4 | IncResV2adv |
|---|---|---|---|---|---|---|---|---|
| ViT-B/16 | PNA+PO | 8.1 | 8.4 | 8.6 | 8.2 | 8.8 | 9.0 | 9.2 |
| | TGR+PO | 7.6 | 8.0 | 8.3 | 7.9 | 8.4 | 8.6 | 8.8 |
| | Ours+SPPO | **6.9**↓ | **7.2**↓ | **7.5**↓ | **7.1**↓ | **7.9**↓ | **7.9**↓ | **8.3**↓ |
| PiT-B | PNA+PO | 7.1 | 7.5 | 7.9 | 7.7 | 8.7 | 8.9 | 9.2 |
| | TGR+PO | 5.9 | 6.4 | 6.8 | 6.5 | 7.7 | 7.8 | 8.4 |
| | Ours+SPPO | **5.0**↓ | **5.4**↓ | **5.9**↓ | **5.6**↓ | **7.1**↓ | **7.3**↓ | **7.9**↓ |
| CaiT-S/24 | PNA+PO | 7.4 | 7.7 | 8.0 | 7.6 | 8.5 | 8.6 | 9.0 |
| | TGR+PO | 6.5 | 7.0 | 7.3 | 6.8 | 7.7 | 7.8 | 8.4 |
| | Ours+SPPO | **5.3**↓ | **5.8**↓ | **6.1**↓ | **5.7**↓ | **6.7**↓ | **6.9**↓ | **7.6**↓ |
| Visformer-S | PNA+PO | 7.0 | 7.2 | 7.8 | 7.4 | 8.7 | 8.9 | 9.3 |
| | TGR+PO | 5.3 | 5.4 | 6.4 | 5.7 | 7.4 | 7.9 | 8.5 |
| | Ours+SPPO | **4.7**↓ | **4.8**↓ | **5.8**↓ | **5.2**↓ | **7.0**↓ | **7.5**↓ | **8.2**↓ |

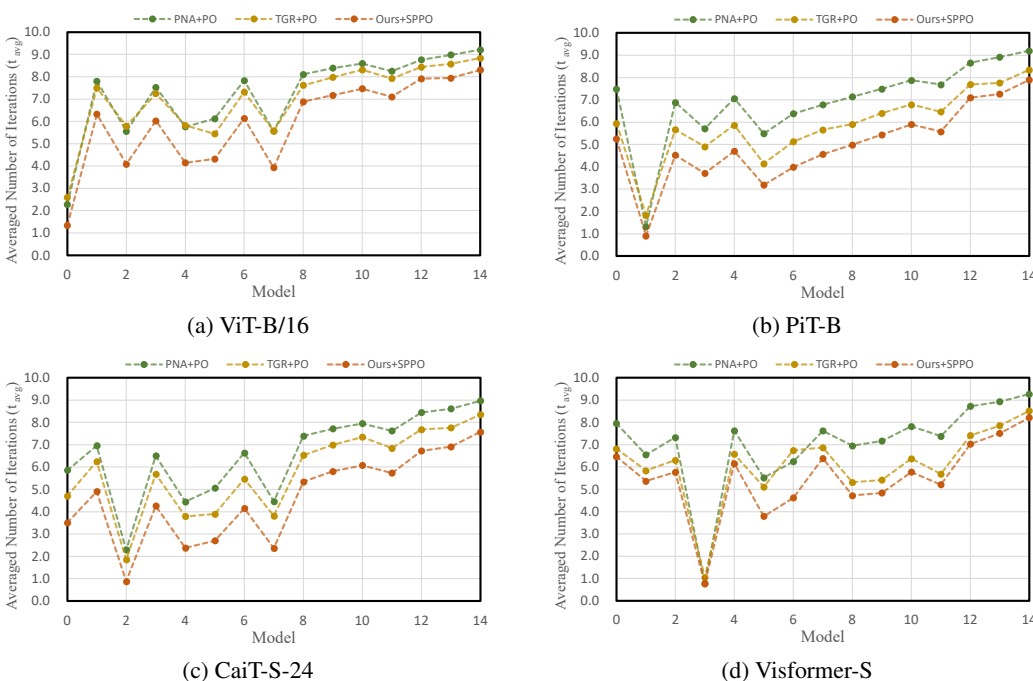

(a) ViT-B/16

(b) PiT-B

(c) CaiT-S-24

(d) Visformer-S

Figure 4: Comparison of attack efficiency of different methods on ViTs, CNNs, and defended CNNs.

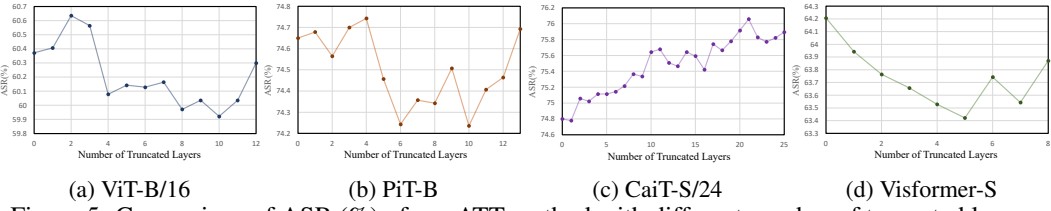

| (a) ViT-B/16 | (b) PiT-B | (c) CaiT-S/24 | (d) Visformer-S |

Figure 5: Comparison of ASR (%) of our ATT method with different number of truncated layers.

## D.3 Additional Analysis of Attack's Effectiveness and Efficiency

We illustrated Fig. 2 more specifically, and in Fig. 6 we visualized class activation maps guided by correct and incorrect labels. It could be clearly seen that, during iterative training, adversarial perturbations gradually allowed the feature information of the correct label to be corrupted, while generating the feature information of the incorrect label. Compared to other methods, our ATT method successfully disrupted the model more effectively (*i.e.*, higher confidence level) and more efficiently (*i.e.*, fewer iteration number). In this figure, it was observed that for the correctly labeled class activation map, the important regions were not exactly in the classification target region, which was obviously a distribution problem incurred by the dataset and the model. This also provoked us to think that a good model should try to avoid classification errors due to background changes, but this did not seem to be achievable on this type of image or model. We believed that a more comprehensive dataset should include the case where the classification target remained the same but the background changed, so that the model could recognize the classification target more accurately.

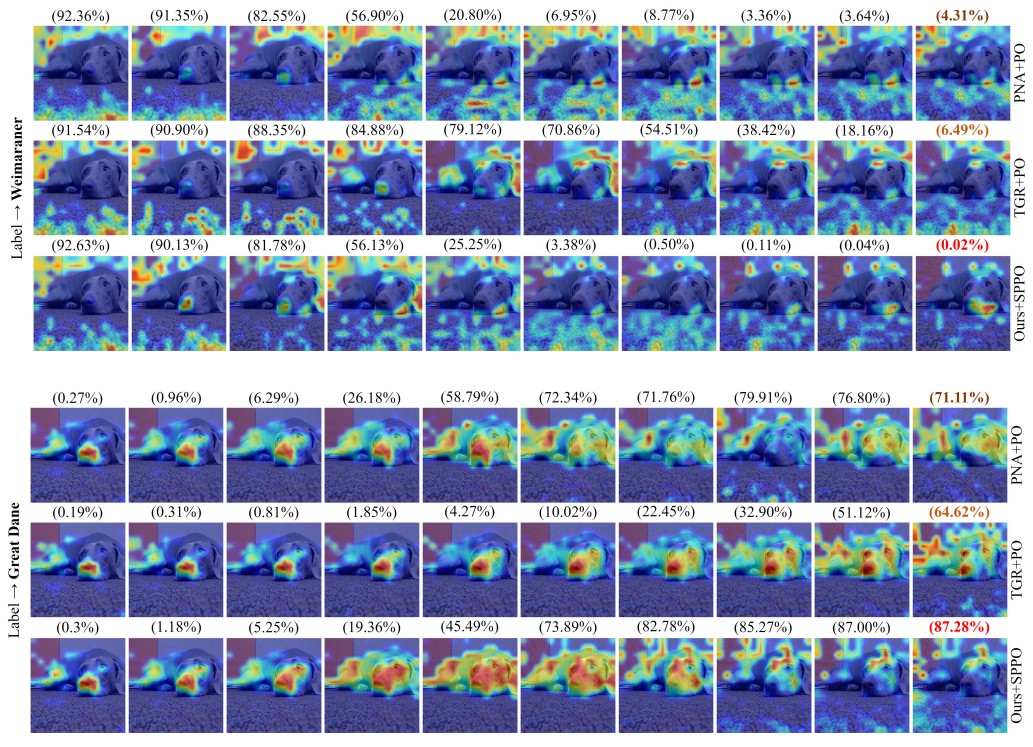

Figure 6: Class activation maps guided by correct label (top) and incorrect label (bottom).

## D.4 Comparison of Class Activation Maps in Intermediate Attack Processes

In Figs. 7, we further showed class activation maps of adversarial examples in intermediate attack processes, trained by different surrogate models. We used a pseudo label (*i.e.*, the label of maximum probability) to guide the generation of class activation maps. Compared to "PNA+PO" and "TGR+PO", our ATT method achieves higher attack efficiency and effectiveness.

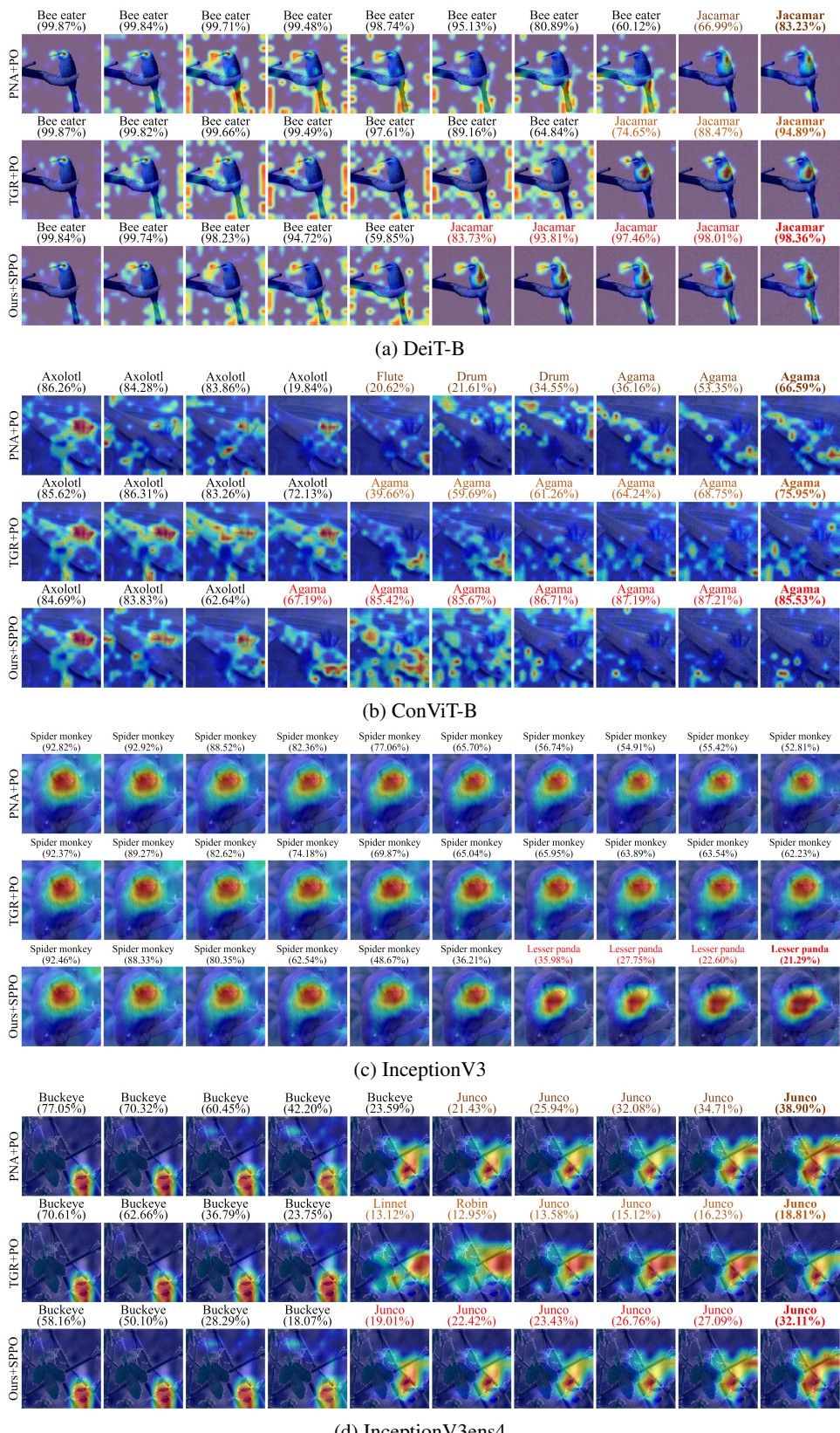

Figure 7: Comparison results of class activation maps in intermediate attack processes under different surrogate models. For each subfigure, the top row is the attack results for "PNA+PO", the middle row is the attack results for "TGR+PO", and the bottom row is the attack results for "Ours+SPPO".

## D.5 The Analysis of Our ATT Attack with Different Module Settings

The analysis results were shown in Tables 7-9, where the ASR is calculated on average for ViTs, CNNs, and defended-CNNs (Def-CNNs) respectively (which will also be used in subsequent experiments). We further conducted analytical experiments on three additional surrogate ViT models to assess the impact of different module settings on attack transferability in black-box scenarios.

In almost all ablation experiments, it was obtained that attacking the Attention module gave the best transferability. But unexpectedly, attacking the MLP module of the CaiT-S-24 model resulted in better transferability. We analyzed that it was due to deeper ViT models like CaiT-S-24, where the MLP module played a greater role in correct classification. Overall, attacking all three modules simultaneously yielded better performance than other module settings in most cases.

Table 7: The average attack success rate (%) of our ATT method with different module settings trained by the surrogate model "PiT-B".

| Attention | QKV | MLP | ViTs | CNNs | Def-CNNs |
|---|---|---|---|---|---|
| - | - | - | 36.8 | 32.0 | 17.3 |
| ✓ | - | - | 70.1 | 58.6 | 33.1 |
| - | ✓ | | 63.9 | 52.6 | 29.0 |
| - | - | ✓ | 60.1 | 47.5 | 26.3 |
| ✓ | ✓ | - | 75.3 | 62.9 | 36.4 |
| ✓ | - | ✓ | 85.0 | 72.2 | 45.1 |
| - | ✓ | ✓ | 79.8 | 68.3 | 43.8 |
| ✓ | ✓ | ✓ | 87.7 | 78.0 | 52.0 |

Table 8: The average attack success rate (%) of our ATT method with different module settings trained by the surrogate model "CaiT-S-24".

| Attention | QKV | MLP | ViTs | CNNs | Def-CNNs |
|---|---|---|---|---|---|
| - | - | - | 69.8 | 43.3 | 27.2 |
| ✓ | - | - | 82.8 | 61.7 | 43.4 |
| - | ✓ | - | 73.2 | 50.0 | 33.9 |
| - | - | ✓ | 85.0 | 62.0 | 45.9 |
| ✓ | ✓ | - | 83.0 | 62.3 | 43.6 |
| ✓ | - | ✓ | 93.0 | 75.6 | 58.3 |
| - | ✓ | ✓ | 86.1 | 64.3 | 47.1 |
| ✓ | ✓ | ✓ | 92.6 | 75.4 | 58.3 |

Table 9: The average attack success rate (%) by our ATT method with different module settings trained by the surrogate model "Visformer-S".

| Attention | QKV | MLP | ViTs | CNNs | Def-CNNs |
|---|---|---|---|---|---|
| - | - | - | 42.5 | 40.8 | 20.6 |
| ✓ | - | - | 73.5 | 67.7 | 35.1 |
| - | ✓ | | 66.5 | 62.5 | 31.8 |
| - | - | ✓ | 62.1 | 64.4 | 35.7 |
| ✓ | ✓ | - | 73.5 | 69.5 | 36.1 |
| ✓ | - | ✓ | 77.1 | 82.3 | 48.5 |
| - | ✓ | ✓ | 72.3 | 77.5 | 44.9 |
| ✓ | ✓ | ✓ | 76.4 | 84.4 | 50.3 |

## D.6 Evaluating the Transferability of Our ATT Attack on Object Detection Models

In the previous experiments, we evaluated the transferability of different attack methods from one classification model to another classification model (*i.e.*, the cross-model transferability). Thus, we further explored the versatility of ViT attacks in cross-task scenarios, where adversarial examples crafted by image classification models (*i.e.*, ViT-B) on the ImageNet dataset were directly input into object detection models. For purpose of better visualization of the cross-task attack process, we used all the intermediate states from 10 updates during the perturbation generation. Additionally, only the top three detection boxes with the highest confidence in the prediction results were considered. We

compared the performance of "Ours+SPPO" with "TGR+PO" on different object detection models, including Faster-RCNN [59], Mask-RCNN [60], SSD [61], RetinaNet [62], and FCOS [63].

As shown in Fig 8, it was clear to see that our ATT attack method allowed the object detection model to produce higher confidence error in detection results.

### D.7 Evaluating the Transferability of Our ATT Attack on Semantic Segmentation Models

To further evaluate the cross-task transferability, we directly input adversarial examples crafted by image classification models (*i.e.*, ViT-B) on the ImageNet dataset into semantic segmentation models. Here, the experimental setup was identical to that used in D.6. We compared the performance of "Ours+SPPO" with "TGR+PO" on different semantic segmentation models, including FCN-resnet101 [64], Deeplabv3-resnet50 [65], and LRASPP [66]. It could be seen in Fig. 9 that our ATT attack method corrupted the segmentation results more severely, which generated more incorrect semantic labels.

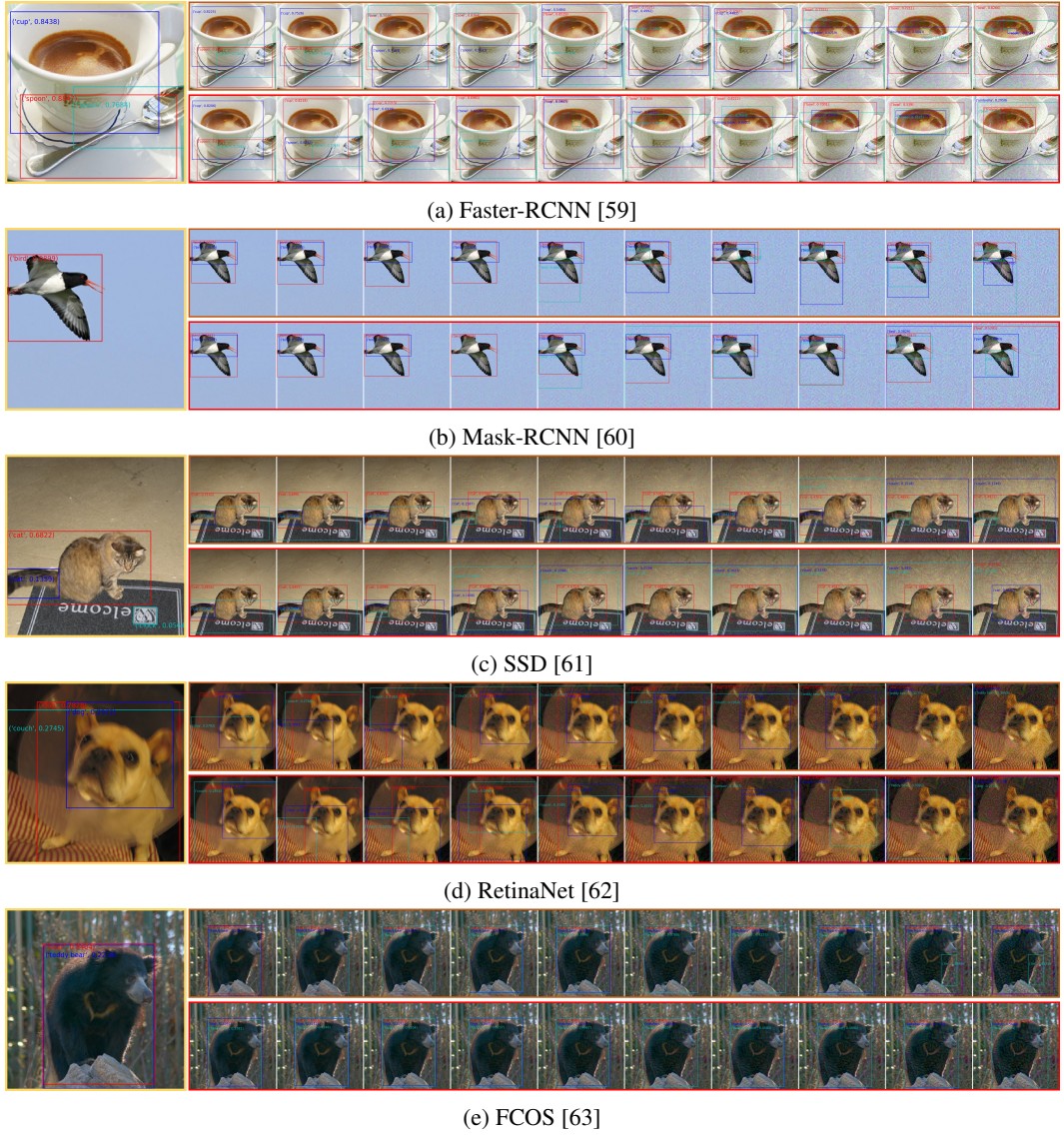

(a) Faster-RCNN [59]

(b) Mask-RCNN [60]

(c) SSD [61]

(d) RetinaNet [62]

(e) FCOS [63]

Figure 8: Demonstration of results of attacking multiple object detection models using adversarial examples trained by ViT-B. For each subfigure, the leftmost shows the detection results for clean images, the top right shows the attack results for "TGR+PO", and the bottom right shows the attack results for "Ours+SPPO".

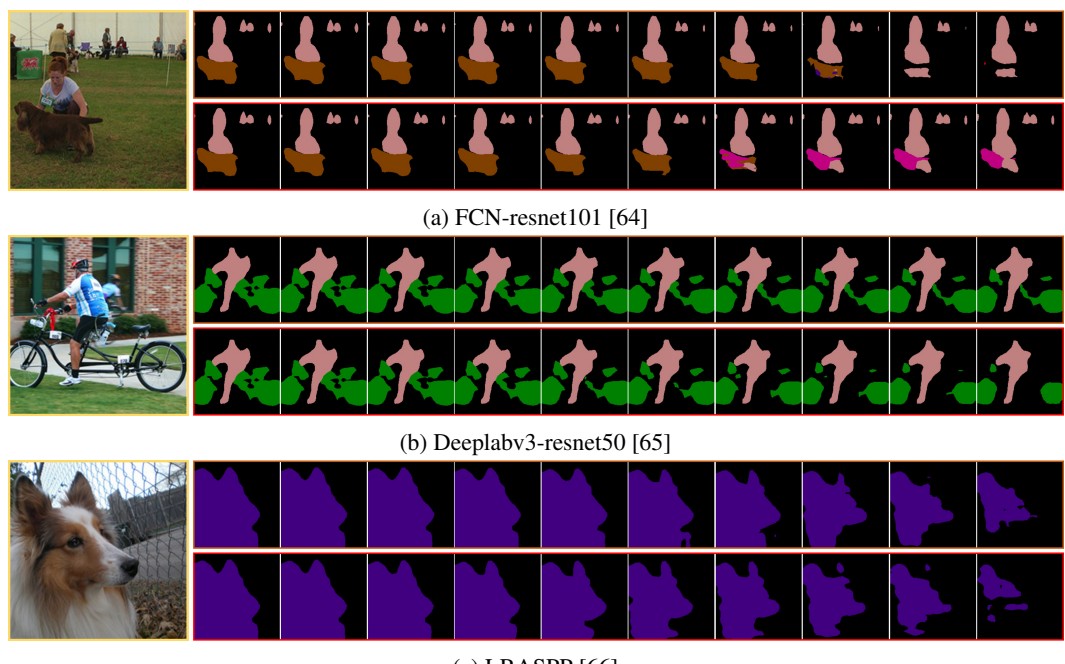

(a) FCN-resnet101 [64]

(b) Deeplabv3-resnet50 [65]

(c) LRASPP [66]

Figure 9: Demonstration of results of attacking multiple semantic segmentation models using adversarial examples trained by ViT-B. For each subfigure, the leftmost is the clean image, the top right is the attack results for "TGR+PO", and the bottom right is the attack results for "Ours+SPPO".

Table 10: Ablation experiments of our ATT method using different attack strategies with the SPPO setup ("AVR": Adaptive Variance Reduced Token Gradient; "ST": Soft Truncation; "HT": Hard Truncation).

| AVR | ST | HT | ViTs | CNNs | Def-CNNs |
|-----|----|----|------|------|----------|
| - | - | - | 49.9 | 29.1 | 19.3 |
| ✓ | - | - | 70.4 | 43.3 | 29.7 |
| - | ✓ | - | 68.8 | 40.7 | 27.7 |
| - | - | ✓ | 67.8 | 39.5 | 28.0 |
| ✓ | ✓ | - | 72.1 | 45.5 | 31.0 |
| ✓ | - | ✓ | 79.3 | 51.7 | 37.5 |
| - | ✓ | ✓ | 77.8 | 48.7 | 35.5 |
| ✓ | ✓ | ✓ | 80.3 | 54.1 | 38.7 |

(a) ViT-B

| AVR | ST | HT | ViTs | CNNs | Def-CNNs |
|-----|----|----|------|------|----------|
| - | - | - | 36.8 | 32.0 | 17.3 |
| ✓ | - | - | 70.1 | 58.6 | 33.1 |
| - | ✓ | - | 63.9 | 52.6 | 29.0 |
| - | - | ✓ | 60.1 | 47.5 | 26.3 |
| ✓ | ✓ | - | 75.3 | 62.9 | 36.4 |
| ✓ | - | ✓ | 85.0 | 72.2 | 45.1 |
| - | ✓ | ✓ | 79.8 | 68.3 | 43.8 |
| ✓ | ✓ | ✓ | 87.7 | 78.0 | 52.0 |

(b) PiT-B

| AVR | ST | HT | ViTs | CNNs | Def-CNNs |
|-----|----|----|------|------|----------|
| - | - | - | 69.8 | 43.3 | 27.2 |
| ✓ | - | - | 82.8 | 61.7 | 43.4 |
| - | ✓ | - | 73.2 | 50.0 | 33.9 |
| - | - | ✓ | 85.0 | 62.0 | 45.9 |
| ✓ | ✓ | - | 83.0 | 62.3 | 43.6 |
| ✓ | - | ✓ | 93.0 | 75.6 | 58.3 |
| - | ✓ | ✓ | 86.1 | 64.3 | 47.1 |
| ✓ | ✓ | ✓ | 92.6 | 75.1 | 58.3 |

(c) CaiT-S/24

| AVR | ST | HT | ViTs | CNNs | Def-CNNs |
|-----|----|----|------|------|----------|
| - | - | - | 42.5 | 40.8 | 20.6 |
| ✓ | - | - | 73.5 | 67.7 | 35.1 |
| - | ✓ | - | 66.5 | 62.5 | 31.8 |
| - | - | ✓ | 62.1 | 64.4 | 35.7 |
| ✓ | ✓ | - | 73.5 | 69.5 | 36.1 |
| ✓ | - | ✓ | 77.1 | 82.3 | 48.5 |
| - | ✓ | ✓ | 72.3 | 77.5 | 44.9 |
| ✓ | ✓ | ✓ | 76.4 | 84.4 | 50.3 |

(d) Visformer-S

## D.8 Additional Ablation Experiments of Our ATT Method using Different Attack Strategies

In Table 10, we performed ablation experiments on three strategies (AVR/ST/HT), where "AVR" denoted Adaptive Variance Reduced Token Gradient, "ST" denoted soft truncation that we utilized the truncation factor to truncate Attention, QKV, and MLP modules, and "HT" denoted hard truncation that we could truncate the gradient of selected Attention layers to zero. It could be seen in Table 10

that "AVR" achieved better attack performances than "ST" and "HT". The hybrid truncation strategy that combined "ST" with "HT" resulted in a higher ASR than using "ST" or "HT" independently. On the whole, when considering all three different attack strategies, our ATT method obtained the best transferability in nearly all of black-box settings.

In Table 11, an additional experiment was conducted to study the variance reduction of token gradients, where both input enhancement and other gradient strategies are not considered here. "Only VR" denoted that only the variance reduction strategy was utilized in our ATT method, while "Only AVR" denoted that only the adaptive variance reduction (AVR) was ultized in our ATT method. As shown in Table 11, our proposed adaptive variance reduction strategy performs better than the fixed variance reduction strategy, leading to better transferability of crafted adversarial examples.

Table 11: Study of the effect of gradient variance reduction on ASR. "Only VR" indicates "Variance Reduced" and "Only AVR" indicates "Adaptive Variance Reduced".

|  | ViT-B/16 | | | PiT-B | | | CaiT-S/24 | | | Visformer-S | | |
|---|---|---|---|---|---|---|---|---|---|---|---|---|
|  | ViTs | CNNs | Def-CNNs | ViTs | CNNs | Def-CNNs | ViTs | CNNs | Def-CNNs | ViTs | CNNs | Def-CNNs |
| Ours(Only VR) | 72.5 | 46.2 | 33.0 | 81.4 | 70.1 | 45.6 | 87.2 | 62.7 | 45.5 | 67.0 | 70.2 | 36.0 |
| Ours(Only AVR) | 73.5 | 47.5 | 33.6 | 84.4 | 72.5 | 46.3 | 89.6 | 66.5 | 47.2 | 67.7 | 72.6 | 37.2 |

## D.9 Evaluating the Transferability of Different Attack Methods on Robust ViT Models

In Table 12, we added a validation experiment for robust ViTs, where "PNA+PO", "TGR+PO", and "Ours+SPPO" use their optimal hyperparameters. "clean" denoted that the model was utilized to classify a clean dataset and the result was the probability of the classifier classifying the data incorrectly. The rest of the attack results were presented by all ASRs. Experimental results showed that our ATT attack method still outperforms SOTA methods in robust ViT models (*i.e.*, DeiT-S [56], Swin-B [57], Xcit-S [58]). Meanwhile, we performed the same test on normal ViT models corresponding to these robust ViT models, where the proposed "Ours+SPPO" also achieved the highest ASR.

Table 12: Comparative experiments of different attack methods on robust ViTs. "clean" indicates that clean images are classified and all results indicate the percentage of classification errors (*i.e.*, ASR).

| Model | Attack | Robust ViTs | | | Normal ViTs | | |
|---|---|---|---|---|---|---|---|
|  |  | DeiT-S | Swin-B | Xcit-S | DeiT-S | Swin-B | Xcit-S |
| clean | | 13.9 | 5.4 | 46.8 | 0.5 | 0.4 | 0.2 |
| ViT-B/16 | PNA+PO | 19.5 | 8.8 | 51.7 | 75.2 | 47.5 | 45.5 |
|  | TGR+PO | 27.7 | 15.8 | 56.5 | 85.1 | 54.4 | 54.5 |
|  | Ours+SPPO | 28.8↑ | 16.9↑ | 56.7↑ | 93.7↑ | 70.4↑ | 68.6↑ |
| PiT-B | PNA+PO | 18.4 | 9.2 | 51.8 | 59.3 | 67 | 71.2 |
|  | TGR+PO | 28.8 | 17.9 | 58.2 | 83.8 | 77.3 | 80.7 |
|  | Ours+SPPO | 29.1↑ | 18.7↑ | 58.3↑ | 91.7↑ | 90.4↑ | 92.8↑ |
| CaiT-S/24 | PNA+PO | 20.3 | 9.9 | 52.2 | 89.5 | 69.3 | 67.1 |
|  | TGR+PO | 32.9 | 20.3 | 56.9 | 96.9 | 76.7 | 77.2 |
|  | Ours+SPPO | 35.5↑ | 21.9↑ | 58.1↑ | 99.2↑ | 90.4↑ | 90.6↑ |
| Visformer-S | PNA+PO | 17.9 | 8.1 | 51.3 | 48.5 | 68.8 | 68.5 |
|  | TGR+PO | 23.6 | 12.9 | 55.8 | 61.5 | 70.6 | 71.8 |
|  | Ours+SPPO | 24.1↑ | 13.8↑ | 56.1↑ | 76.5↑ | 86.5↑ | 86.7↑ |

