# OpenReview forum: "Boosting the Transferability of Adversarial Attack on Vision Transformer with Adaptive Token Tuning"
_NeurIPS.cc/2024/Conference — NeurIPS 2024 poster_

### Official Review · Reviewer_52P8 · 2024-07-11

**Soundness:** 3
**Presentation:** 3
**Contribution:** 3
**Rating:** 6
**Confidence:** 4

**Summary:**

This paper introduces Adaptive Token Tuning (ATT) to improve the transferability of adversarial examples generated from ViTs. ATT is an improvement over traditional gradient-based algorithms, consisting of three independent methods. The first method reduces gradient variance by rescaling the token gradients computed throughout different modules and layers in the ViT. The second method is a scheduled, semantic-guided patch-out approach to increase the diversity of input during the iterative gradient ascent procedure. The third method is a truncation approach for attenuating the model-specific attention occurring at the deeper layers of the ViT. Experimental results on ImageNet show an improvement over existing transfer-based attacks.

**Strengths:**

**Originality:**

The proposed ATT algorithm includes three strategies that work in combination to improve the transferability of perturbations generated from vision transformers. Each method is developed based on insights from previous work and improves upon them. The proposed method is original.


**Quality/Clarity:**

The presentation is clear, and the methods are well-explained. A thorough evaluation of transferability on a variety of target models, including transformers and both undefended and defended CNNs.


**Significance:**

The proposed work is a valuable addition to methods for generating more transferable perturbations from vision transformers.

**Weaknesses:**

1. Ln76: Blackbox attack also includes those when target model has limited access, i.e., query-based attacks.

2. The statement in ln 294-295 requires further consideration. The analysis itself does not show that the ATT perturbation captured more features than those generated by other algorithms. While the improvement suggests this, there is no analysis proving the statement.

3. Several statements in the paper requires references, for instance, Ln218-219.

**Questions:**

1. Do we have results showing the benefit of the adaptive strategy in variance reduction in terms of transferability improvement?

2. In (2), is it possible for the scaling factor to be negative? Wouldn't that change the direction of the gradient, resulting in a less optimal argmax of (1), and potentially making the perturbation less transferable?

3. For the Hybrid Token Gradient Truncation Method, many design decisions are made without clear explanation. For example, why do we have two different types of truncations? I appreciate the authors for including the analysis in A.5 on the number of truncated layers. For the experiment in 4.1, how is $\ell'$ selected for each model? Do we need to run a sweep for each model?

4. For the results in Tables 1 and 2, do all methods, including the baselines and ours, have Patchout?

5. In Ln 271, shouldn't the number of discards be equal between Patchout and SPPO for a fair comparison?

**Limitations:**

No discussion on limitation of the work. The checklist says the limitation is discussed in Sec 2 4 and Appendix A1, but I could not find any discussion.

---

> ### Author Rebuttal · Authors · 2024-08-06
>
> 1. **Answer to Weakness 1 -- query-based attacks:** We appreciate your comment on the broader definition of blackbox attacks, including query-based scenarios where the target model has limited accessibility. We will expand our discussion to encompass these types of attacks, providing a more comprehensive overview of blackbox scenarios in the context of adversarial machine learning. This will be incorporated in the introduction and related work sections to clarify the scope and applicability of our Adaptive Token Tuning (ATT) method.
>
> 2. **Answer to Weakness 2 -- the ATT perturbation captured more features:** We acknowledge the reviewer's point regarding the need for a more robust analysis to substantiate our claim that ATT perturbations capture more feature information. Referred to our explanation of **Questions in review 9KoY**. This could be interpreted as a further explanatory illustration of our method from the perspective of experimental results, which could retain more feature information to improve the transferability of the adversarial attack. Additionally, in response, we will refine our experimental section to include a comparative feature analysis between ATT and other algorithms. This analysis will aim to quantitatively demonstrate how ATT retains more relevant features, thereby enhancing transferability. This addition will address the gap between our observational claims and the empirical evidence provided.
>
> 3. **Answer to Weakness 3 -- References required for Ln218-219:** Thank you for highlighting the need for additional references in our manuscript. We recognize the importance of properly citing existing work to validate our statements. Specifically, for the assertions made in Ln218-219, we will ensure that all relevant literature is cited, providing a solid foundation for our claims and aligning with academic standards.
>
> 4. **Answer to Questions 1 -- Adaptive Variance Reduced and transferability:** We appreciate the inquiry into the impact of our adaptive strategy on variance reduction. As detailed in Table 4 of the supplementary PDF, our experiments demonstrate two key aspects: **(1)** variance reduction while preserving feature information, and **(2)** adaptive variance reduction that smooths gradient variance across layers.
> Our analysis in Supplementary A.2 confirms that our strategy effectively reduces gradient variance, optimizing $\lambda$ to balance the trade-off between fixed and overly penalized scaling, which could hinder the transferability. This optimal setting ensures improved attack efficacy without compromising the adaptive nature of the method.
>
> 5. **Answer to Questions 2 -- The scaling factor to be negative?:** In our implementation (referenced in Line 147, as well as the code implementation), we ensure that the scaling factor remains non-negative, thus maintaining the original direction of token gradients. This is crucial for sustaining an effective adversarial attack, as negative values would indeed invert the gradient direction, undermining the attack's efficacy.
>
> 6. **Answer to Questions 3 -- Hybrid Token Gradient Truncation Method:** The Hybrid Token Gradient Truncation Method is designed to optimize the utilization of the attention mechanism by adjusting gradients in different Our motivation, grounded in literature and empirical evidence (see references [16] in our submitted manuscript), seeks to understand and enhance the impact of different modules in ViT layers on the perturbation transferability.
> For each surrogate model tested, we need to sweep them all, but parameters are fixed except for those under examination, facilitating focused analysis as described in Supplementary A.5.
>
> 7. **Answer to Questions 4 -- Do Tables 1 and 2 include PatchOut?:** Regarding input enhancement strategies, Tables 1 and 2 deliberately exclude PO/SPPO to isolate the effects of our adaptive gradient and truncation strategies from those of input modifications. This choice was made to clearly demonstrate the independent effectiveness of our proposed methods.
>
> 8. **Answer to Questions 5 -- Number of PatchOut:** Our Self-Paced Patch Out (SPPO) approach dynamically adjusts the number of discards, unlike the fixed value used in Patch Out (PO). Since more discards reduce the features that perturbations can learn, we ensure a fair comparison by setting the expected number of discards in SPPO to match or exceed that of PO.
>
> 9. **Answer to Limitations:** We appreciate the reviewer's attention to the limitations of our work. We realize there was an oversight in our citation of the sections discussing limitations, which may have caused confusion. To address this, we will add a dedicated section in our manuscript that explicitly outlines the limitations of our Adaptive Token Tuning (ATT) method. This new section will discuss potential scalability issues, the dependency on hyperparameters, and any constraints related to the types of adversarial settings where our method may not perform optimally. We aim to provide a comprehensive and transparent overview of these aspects to better inform future research and application of our findings.

---

> > ### Comment · Reviewer_52P8 · 2024-08-12
> >
> > Thank you for the responses. Most of my concerns have been address. For **Answer to Weakness 2 -- the ATT perturbation captured more features**, I strongly suggest that the authors either include additional experiments to support this statement or provide more specificity in the original claim. For **Answer to Questions 2 -- The scaling factor to be negative?**, I also suggest clarifying this in the section.

---

> > > ### Author Response · Authors · 2024-08-13
> > >
> > > Dear Reviewer 52P8:
> > >
> > > Thank you for your suggestions on the various aspects of our work. We will provide a more accurate and detailed description of our methodology in our revised manuscript. We also appreciate your high recognition of our research.

---

### Official Review · Reviewer_38RD · 2024-07-11

**Soundness:** 3
**Presentation:** 3
**Contribution:** 3
**Rating:** 5
**Confidence:** 2

**Summary:**

The paper proposes an Adaptive Token Tuning (ATT) method to enhance the transferability of adversarial attacks on Vision Transformers (ViTs). The method introduces three optimization strategies: adaptive gradient re-scaling to reduce token gradient variance, a self-paced patch out strategy to enhance input diversity, and a hybrid token gradient truncation strategy to reduce the effectiveness of the attention mechanism. Extensive experiments demonstrate the superiority of ATT over existing methods in terms of attack success rate and transferability across various models.

**Strengths:**

1. The combination of adaptive gradient re-scaling, self-paced patch out, and hybrid token gradient truncation is novel and well-motivated, providing a fresh perspective on improving adversarial attack transferability for ViTs.
2. The experimental results show that ATT significantly outperforms state-of-the-art methods, achieving higher attack success rates and better transferability.

**Weaknesses:**

1. In the paper's formulation, there are multiple hyperparameters. Do these hyperparameters interact with each other? Why is γ set to 0.5?
2. In Figure 2, is the model used for prediction the same as the target model for the attack? From the figure, it appears that the proposed method indeed makes the model focus on the classification target more quickly. However, how do you explain that as the number of iterations increases, the attention no longer focuses on the classification target? The targets shown in the figure are all dogs, just different types, so the attention should still focus on the dog.
3. Have the effects been tested on defended ViTs?

**Questions:**

Refer to Weakness section.

**Limitations:**

No.

---

> ### Author Rebuttal · Authors · 2024-08-06
>
> 1.  **Answer to Weakness 1 -- Hyperparameters Interaction:**
> The hyperparameters in our Adaptive Token Tuning (ATT) method fall into two categories: those related to gradient adjustment and those pertaining to input enhancement.
> To mitigate the interaction between hyperparameters,
> these categories are adjusted independently and alternatively.
> For the gradient adjustment, the gradient penalty factor $\gamma$ is initially set to 0.5 as a median value to mildly rescale back-propagated token gradients. It will be adaptively adjusted within the ranges of $(0, 0.5]$ and $[0.5, 1)$ based on the gradient variance across consecutive ViT layers.
> Based on our exploratory experiments, this value of $\gamma$ was not only a practical but also an effective choice.
>
> 2.  **Answer to Weakness 2 -- Model Prediction and Target Focus in Figure 2:**
> In our experiments (see Ln333-336), ViT-B served as the surrogate model, with visualizations in Figure 2 conducted on the black-box model CaiT-S/24 using Grad-CAM. The visualization does not specifically set a label; thus, attention is drawn to the category deemed most probable by the model. The shift in focus observed in the figure illustrates a change in the highest probability category, indicating a successful attack, characterized by a rapid and confident misclassification. This dynamic is further elaborated in Figure 6, where attention visualizations are presented for both correct and incorrect labels, highlighting the mechanism of our attack’s effectiveness.
>
> 3.  **Answer to Weakness 3 -- Testing on Defended ViTs:**
> As addressed in response to **Weaknesses 2 from Review zV8Q**, we have expanded our experimental validation to include defended (robust) ViTs, with findings detailed in **Table 3 of the rebuttal PDF**.
> This additional testing confirms the effectiveness of our proposed ATT attack against various adversarial defenses, ensuring that our approach is not only theoretically sound but also practically viable against contemporary robust architectures.

---

> > ### Comment · Reviewer_38RD · 2024-08-12
> >
> > My concerns are properly addressed.
> >
> > Thanks for the rebuttal!

---

> > > ### Author Response · Authors · 2024-08-13
> > >
> > > Dear Reviewer 38RD:
> > >
> > > Thank you for your suggestions regarding the details of our experiments and broader experimental comparisons. We appreciate your feedback and the acknowledgment of the clarifications and new experiments, as well as your positive recognition of our work.

---

### Official Review · Reviewer_zV8Q · 2024-07-11

**Soundness:** 3
**Presentation:** 3
**Contribution:** 2
**Rating:** 4
**Confidence:** 4

**Summary:**

n this paper, the authors investigate three strategies to boot the transferability of adversarial attacks on Vision Transformers, including an adaptive gradient re-scaling strategy, a self-paced patch out strategy, a hybrid token gradient truncation.

**Strengths:**

1 The experiments are solid.

2 The soundness of the method is good.

3 This paper is well written.

**Weaknesses:**

1 The title of the  paper might be imappropriate to conclude the techniques proposed by authors. The overall methods are composed of three partitions:

- An adaptive gradient re-scaling strategy to reduce the overall variance of token gradients. It is related to the back propagation for crafting adversarial samples.

- A self-paced patch out strategy to enhance the diversity of input tokens. It is related to the adaptive forward propagation for calculating the adversarial loss.
- A hybrid token gradient truncation strategy to weaken the effectiveness of attention mechanism. It is also related to the back propagation for crafting adversarial perturbations.

We can see that none of these points are related to the token tuning. (Maybe I am wrong, I hope authors can point out my mistakes.) As I understand it, token tuning usually refers to optimizing the token after tokenizing the input images.  Since the designed attacks craft pertubations directly on input images like most of previous attacks. I think it might be imappropriate to conclude the proposed methods with the phrase "Adaptive Token Tuning".

2 In line 72, the authors claim that "robustly trained defense models still fail to meet security requirements".  The observation contradicts with the findings observed in [1].  I conjecture this is because the authors choose to attack secure models which is published in 6 years ago. Considering the rapid development in the adversarial community, I recommend to attack recent sucure models trained by $l_{\infty}$ AT provided by robustbench [2].

3 The novelty of this paper is fair. The design principles of the proposed methods originates from previous papers and in this paper, they simply make them adaptive. Higher flexibility often brings better transferability. However, it also brings additional costs: it will increases the difficulty of hyperparameter tuning. In addition, more new insights are needed to guide community better leverage ViT to attack deep models.

4  The conparison with the baseline methods in Table 1 and Table 2 might be unfair. It seems that the proposed methods acquire better transferability than other attacks. However, considering the proposed attack proposed in this paper combines the power of three attacks: TGR, PNA, PatchOut. I think the appropriate baseline is TGR+PNA+PatchOut. It helps us more precisely pinpoint the gains brought by the proposed method.

**Questions:**

In Table 2, the results reveal that the proposed method can better attack the robust CNNs. How about robust ViTs, such as DiT-S in [3], Swin-B in [4], Xcit-S in [5]?

**Limitations:**

There is no single section to discuss the limitation and the broader Impact of the paper. Can you explain them with more details?

[1] https://github.com/Trustworthy-AI-Group/TransferAttacks

[2] https://robustbench.github.io/

[3] Are Transformers More Robust Than CNNs? in NeurIPS 2021.

[4] When Adversarial Training Meets Vision Transformers: Recipes from Training to Architecture, in NeurIPS 2022.

[5] A Light Recipe to Train Robust Vision Transformers, in SaTML 2023.

---

> ### Author Rebuttal · Authors · 2024-08-06
>
> 1. **Answer to Weakness 1 -- Adaptive Token Tuning for our manuscript title:** We appreciate the reviewer's emphasis on the traditional definition of token tuning. In our work, token tuning extends beyond mere feature value processing; it encompasses the modification of feature values, attention weights, gradient information, and layer interaction, all centered around the token as the basic unit.
> Our strategies, whether related to gradient tuning or input enhancement, adhere to this broader definition of token tuning. We will clarify this in our revised manuscript to avoid any misunderstanding.
>
> 2. **Answer to Weakness 2 -- Testing on Defended ViTs:** Based on the reviewer's suggestion, we conducted additional experiments on robust ViTs, as detailed in **Table 3 of our rebuttal PDF**.
> Transferable black-box attacks were tested against three robust ViT models and their normal counterparts using our best strategies and those of comparative methods.
> The results confirm that our approach consistently outperforms others in effectiveness.
>
> 3. **Answer to Weakness 3 -- Description of our innovations:**
> **(1)** Our gradient treatment strategy, although inspired by existing methods like PNA and TGR, we proposed a new treatment strategy for the gradient, and that the idea of treating the gradient was completely different from PNA and TGR, see our manuscript Ln146-154. On this basis, introduced a novel adaptive adjustment strategy that aligned gradients from previous layers with those of the deepest layer. This alignment helped smooth out gradient variance and enhanced feature information learning, moving beyond mere adaptivity.
> **(2)** Our self-paced patch out strategy innovates beyond the random discard of patches. Our strategy introduced the concept of self-stepping learning and meanwhile distinguished between significant and non-significant regions when discarding tokens. This strategy is beneficial for learning the transferable attack pattern and also improves the efficiency of the attack, as demonstrated in Figures 2, 4, and Table 5 of our manuscript.
> The relationship between our method and PatchOut is more like the relationship between dropout and its variants.
> **(3)** Unlike TGR and PNA, our Hybrid Token Gradient Truncation truncates attention modules in deep ViT layers to learn generic feature patterns and adjusts corresponding truncation factors based on the role of different modules.
> This nuanced approach differs significantly from previous methods and offers a tailored strategy for enhancing attack transferability.
> In sum, the above-mentioned strategies are different from the existing ViT gradient attacks. Our approach was a **`new`** and **`innovative`** approach to attacks on gradients.
>
> 4. **Answer to Weakness 4 -- Tables 1-2 were fair:** We apologize for any ambiguity in our manuscript regarding the experimental setups described in Tables 1-3.
> The explanation of the experiment’s fairness can be found in our responses to **Weaknesses 1 and 2 from Reviewer 9KoY**.
> Regarding the combination of TGR, PNA, and PatchOut, our experimental design in Table 3 of our manuscript considers the synergistic and antagonistic effects of combining these methods.
> We found that simply superimposing TGR and PNA could lead to excessive feature information loss, contradicting the suggested "PNA+TGR" approach.
> An additional comparison experiment is provided in **Table 1 of our rebuttal PDF**.
> Patchout could be used as an enhancement input, so in the comparison experiments, we made such a comparison in our manuscript Table 3.
>
> 5. **Answer to Questions -- Testing on Defended ViTs:**
> To address this query, we extended our experiments to include robust ViT models like DiT-S, Swin-B, and Xcit-S. These results are presented in the updated **Table 3 in our rebuttal PDF**.
> Our method shows comparable effectiveness against these robust ViTs, validating the adaptability and efficiency of our strategies across different robust architectures.
> This inclusion provides a comprehensive view of our method's performance against state-of-the-art robust ViTs, in line with current advancements in adversarial machine learning.
>
> 6. **Answer to Limitations:**
> We thank the reviewer for paying attention to the limitations of our work. We realize that there was an omission in our citation of the sections discussing limitations, which might have led to confusion. To handle this, we will add a dedicated section in our manuscript to clearly outline the limitations of our Adaptive Token Tuning (ATT) method. This new section will discuss potential scalability issues, the dependence on hyperparameters, and any constraints related to the types of adversarial settings where our method may not perform at its best. We will also take into consideration the social implications. We aim to provide a comprehensive and transparent overview of these aspects to better inform future research and application of our findings.

---

> ### Comment · Reviewer_zV8Q · 2024-08-12
>
> Thank you for your detailed rebuttal. It indeed addresses most of my concerns. However, the gain in Table 1 of your rebuttal indicates that it is so marginal for me, Thus I will only increase my score marginally to 4.

---

> > ### Author Response · Authors · 2024-08-13
> >
> > Thank you for considering our clarifications and additional experiments in the rebuttal. We greatly appreciate the valuable review and your decision to increase the rating. We will incorporate the discussions to our revised manuscript accordingly.
> >
> > In response to your concern about Table 1 in our rebuttal PDF, we would like to address any potential ambiguity and provide further clarification on our experimental results from three perspectives.
> >
> > In all the following tables, we consider four ViTs (ViTB/16, PiT-B/16, CaiT-S/24, Visformer-S) as surrogate models. The generated adversarial examples are then used to attack black-box target models, and the average ASR is calculated separately for 8 ViT models, 4 CNN models, 3 Adversarially Trained Defense CNN (a.k.a., Def-CNNs) models. Here, we primarily focus on comparing our proposed ATT attack method with two state-of-the-art methods, PNA [15] and TGR [14].
> >
> > **(1)** Attacking ViTs via "Token Gradient-based Optimization":
> > Firstly, we focus solely on the ViT attack using "Token Gradient-Based Optimization." We denote these methods with `(w/o)` to indicate that it is "without input diversity enhancements."
> >
> > **Tables A**
> > | Model | Attack | ViTs | CNNs | Def-CNNs | -- | Model | Attack | ViTs | CNNs | Def-CNNs |
> > |--|--|--|--|--|--|--|--|--|--|--|
> > |  | PNA(w/o) | 62.8 | 38.8 | 26.3 | |  | PNA(w/o) | 63.9 | 54.6 | 30.3 |
> > | ViT-B/16 | TGR(w/o) | 69.8 | 42.7 | 29.3 | | PiT-B/16 | TGR(w/o) | 78.7 | 68.0 | 39.3 |
> > |  | ATT(w/o) | **77.4`+7.6`** | **49.8`+7.1`** | **36.0`+6.7`** | |  | ATT(w/o) | **85.9`+7.2`** | **75.3`+7.3`** | **48.0`+8.7`** |
> > | -- | -- | -- | -- | -- |--| -- | -- | -- | -- | -- |
> > |  | PNA(w/o) | 79.2 | 52.1 | 34.9 | |  | PNA(w/o) | 54.9 | 52.1 | 25.5 |
> > | CaiT-S/24 | TGR(w/o) | 84.8 | 54.0 | 36.1 | | Visformer-S | TGR(w/o) | 62.7 | 62.2 | 30.6 |
> > |  | ATT(w/o) | **91.2`+6.4`** | **68.2`+14.2`** | **50.2`+14.1`** | |  | ATT(w/o) | **67.0`+4.3`** | **77.1`+14.9`** | **41.1`+10.5`** |
> >
> > Specifically, our method ATT(w/o) employs two strategies: adaptive variance reduced token gradient and hybrid token gradient truncation. PNA(w/o) uses attentional skipping without any input diversity enhancement. TGR(w/o) relies solely on use gradient regularization, also without input diversity enhancement.
> >
> > Compared to state-of-the-art methods without input diversity enhancement, our ATT(w/o) shows a significant improvement in attack performance, with the increase in the average ASR highlighted by “`+∆`” in the Table A.
> >
> > **(2)** Attacking ViTs via "Token Gradient-based Optimization" and " Input Diversity Enhancement":
> > Secondly, we incorporate the "Input Diversity Enhancement" strategy with "Token Gradient-based Optimization", where Patch Out (PO) introduced in PNA [15] is used as the input diversity enhancement strategy. As a result, we denote these methods with `(PO)` to indicate that it is "with patch out-based input diversity enhancement."
> >
> > **Tables B**
> > | Model | Attack | ViTs | CNNs | Def-CNNs | -- | Model | Attack | ViTs | CNNs | Def-CNNs |
> > |--|--|--|--|--|--|--|--|--|--|--|
> > |  | PNA(PO) | 70.8 | 42.6 | 29.9 | |  | PNA(PO) | 73.1 | 57.8 | 32.7 |
> > | ViT-B/16 | TGR(PO) | 76.0 | 46.7 | 33.3 | | PiT-B/16 | TGR(PO) | 82.3 | 68.9 | 41.3 |
> > |  | ATT(PO) | **77.1`+1.1`** | **51.7`+5.0`** | **37.1`+3.8`** | |  | ATT(PO) | **84.2`+1.9`** | **75.2`+6.3`** | **48.4`+7.1`** |
> > | -- | -- | -- | -- | -- |--| -- | -- | -- | -- | -- |
> > |  | PNA(PO) | 81.6 | 56.6 | 39.3 | |  | PNA(PO) | 68.8 | 61.8 | 32.3 |
> > | CaiT-S/24 | TGR(PO) | 88.8 | 60.5 | 40.5 | | Visformer-S | TGR(PO) | 70.4 | 64.3 | 33.5 |
> > |  | ATT(PO) | **91.1`+2.3`** | **71.9`+11.4`** | **54.3`+13.8`** | |  | ATT(PO) | **70.5`+0.1`** | **79.3`+15.0`** | **44.5`+11.0`** |
> >
> > Specifically, our ATT(PO) method employs three strategies: adaptive variance-reduced token gradient, hybrid token gradient truncation, and patch out. On the other hand, PNA(PO) and TGR(PO) integrate patch out with their respective token gradient-based optimizations. It’s worth noting that TGR(PO) is currently the state-of-the-art method for ViT attacks. Additionally, the review suggested TGR+PNA+PO as a baseline, while the differing token gradient mechanisms used by TGR and PNA present a conflict.
> >
> > It needs to be emphasized that our ATT(PO) method is not the full version for this comparison in Table B, as it uses the same Patch Out enhancement as PNA(PO) and TGR(PO). The Table B above highlights the average ASR improvement of our ATT(PO) method compared to state-of-the-art methods with input diversity enhancement, indicated by “`+∆`”.
> >
> > **(3)**...

---

> > > ### Author Response · Authors · 2024-08-13
> > >
> > > **(3)** Attacking ViTs via "Token Gradient-based Optimization" and "Improved Input Diversity Enhancement":
> > > Thirdly, we further investigate the ViT attack strategy that combines "Token Gradient-based Optimization" and "Improved Input Diversity Enhancement", where our proposed Self-Paced Patch Out (SPPO) is used as the improved input diversity enhancement strategy. As a result, we denote these methods with `(SPPO)` to indicate that it is "with self-paced patch out-based input diversity enhancement."
> > >
> > > **Tables C**
> > > | Model | Attack | ViTs | CNNs | Def-CNNs | -- | Model | Attack | ViTs | CNNs | Def-CNNs |
> > > |--|--|--|--|--|--|--|--|--|--|--|
> > > | ViT-B/16 | TGR(SPPO) | 71.6 | 47.4 | 33.1 | | PiT-B/16 | TGR(SPPO) | 81.6 | 72.4 | 47.4 |
> > > |  | ATT(SPPO) | **80.3`+8.7`** | **54.1`+6.7`** | **38.7`+5.6`** | |  | ATT(SPPO) | **87.7`+6.1`** | **78.0`+5.6`** | **52.0`+4.6`** |
> > > | -- | -- | -- | -- | -- |--| -- | -- | -- | -- | -- |
> > > | CaiT-S/24 | TGR(SPPO) | 86.4 | 64.9 | 48.4 | | Visformer-S | TGR(SPPO) | 65.4 | 80.9 | 45.4 |
> > > |  | ATT(SPPO) | **92.6`+6.2`** | **75.4`+10.5`** | **58.3`+9.9`** | |  | ATT(SPPO) | **76.4`+11.0`** | **84.4`+3.5`** | **50.3`+4.9`** |
> > >
> > > In this comparison, our ATT(SPPO) method represents the full version, incorporating adaptive variance-reduced token gradient, hybrid token gradient truncation, and self-paced patch out. Additionally, we replaced the original Patch Out strategy used in TGR with our SPPO strategy, resulting in and TGR(SPPO).
> > >
> > > As shown in Tables A, B, and C, our ATT(SPPO) method consistently outperforms across various attack scenarios. Table C also demonstrates that TGR(SPPO) further improves the average ASR compared to TGR(PO) in Table B, highlighting the effectiveness and compatibility of our proposed SPPO strategy.
> > >
> > > Thank you again for considering our work. We hope this explanation addresses your concern.

---

### Official Review · Reviewer_9KoY · 2024-07-11

**Soundness:** 3
**Presentation:** 3
**Contribution:** 2
**Rating:** 5
**Confidence:** 4

**Summary:**

This paper investigates the enhancement of adversarial attack transferability on Vision Transformers (ViTs) through innovative adaptive token tuning techniques. It addresses the vulnerability of ViTs to adversarial attacks by introducing three main optimization strategies: an adaptive gradient re-scaling method to uniformly reduce gradient variance across ViT layers, a self-paced patch out strategy to increase input diversity and mitigate overfitting by dynamically discarding less important perturbation patches, and a hybrid token gradient truncation method designed to attenuate the attention mechanism's effectiveness, adjusting truncation factors across different modules for optimal balance. Extensive experiments demonstrate that these methods not only improve the transferability of adversarial examples across ViTs and CNNs but also significantly enhance attack success rates by an average of 10.1% compared to state-of-the-art transfer-based attacks. Notably, this approach achieves a remarkable average attack performance of 58.3% on defended CNNs, highlighting the ongoing challenges in securing robustly trained defense models against sophisticated adversarial strategies.

**Strengths:**

1. This paper significantly enhances the transferability of adversarial attacks on Vision Transformers (ViTs) by introducing three main optimization strategies: an adaptive gradient re-scaling method, a self-paced patch out strategy, and a hybrid token gradient truncation method. It also provides a comprehensive theoretical analysis of the proposed methods, detailing the mechanisms through which each contributes to improving adversarial robustness and effectiveness.

2. Extensive experimental results substantiate the effectiveness of the proposed methods, demonstrating their efficacy across various settings and models.

**Weaknesses:**

1. This paper enhances the transferability of adversarial attacks on Vision Transformers (ViTs) through adaptive token tuning, introducing three main optimization strategies: an adaptive gradient re-scaling method, a self-paced patch out strategy, and a hybrid token gradient truncation method. However, it lacks ablation experiments for these three strategies.

2. The authors propose an improvement on the PatchOut method, termed Self-Paced Patch Out, which is an input transformation approach that intuitively could enhance adversarial transferability. However, comparing it in Table 1 alongside methods that do not incorporate input transformation, such as TGR, seems unfair.

**Questions:**

The authors mention that the mild re-scaling strategy can back-propagate some important feature information in the largest token gradient, which is correlated with important feature information, unlike TGR. However, it is unclear how retaining important feature information contributes to enhanced transferability of adversarial attacks. Could the authors elaborate on the connection between these retained features and the observed improvements in transferability?

**Limitations:**

The authors suggest that attention modules in certain ViT layers are redundant for image classification and adversarial perturbation generation, potentially leading to overfitting. This assertion raises questions about the underlying mechanisms. Could the authors provide more detailed analysis or empirical evidence on how these redundant attention modules affect the transferability of adversarial attacks?

---

> ### Author Rebuttal · Authors · 2024-08-06
>
> 1. **Answer to Weakness 1 -- ablation experiments for these three strategies:** We apologize for any ambiguity in our manuscript regarding the experimental setups described in Tables 1-3, where we did an ablation study of the input enhancement strategy including both Path Out (PO) and Self-Paced Patch Out (SPPO).
> To clarify, the input enhancement strategy (PO and SPPO) was not included in Tables 1-2, aligning with the setups used for TGR.
> In Table 3, the input enhancement strategy is combined with our other two attack strategies.
> Additionally, we provided the ablation experiment of the other two proposed strategies in **Table 2 of our rebuttal PDF**, including the adaptive gradient re-scaling method (see AVR) and the hybrid token gradient truncation method (see ST and HT).
>
> 2. **Answer to Weakness 2 -- Table 1-2 were fair:**
> To align with the setups used for TGR, we did not include any input enhancement strategies in Tables 1-2, ensuring a fair comparison between our ATT method and the state-of-the-art methods.
> On the other hand, we investigated the effect of SPPO combined with other methods (such as TGR and PNA) on attack transferability in **Table 1 of our rebuttal PDF**, further verifying the effectiveness of our proposed SPPO strategy.
>
> 3. **Answer to Questions -- Retention of features to improved transferability:** The retention of important feature information is pivotal for the effectiveness of gradient-based adversarial attacks, which often utilize the gradient direction opposite to these critical features to enhance attack transferability. While completely disregarding these features, as seen with TGR, reduces overfitting, it also diminishes the utility of these features in facilitating the transferability of perturbations.
> Our experiments, detailed in **Table 4 of the rebuttal PDF**, specifically explore the impact of preserving these important features using our methodology—without truncation or input enhancement strategies—demonstrating their significance in improving adversarial attack transferability.
>
> 4. **Answer to Limitations -- Redundant attention modules affected the transferability:** To address concerns regarding the redundancy of attention modules, we provide a comprehensive analysis in Section A.5 of our Supplementary Material.
> This section includes experimental evidence showing how excessive reliance on the attention mechanism can lead to overfitting, thus negatively impacting the transferability of adversarial attacks.
> By truncating the attention layer, we demonstrate the potential for improved attack transferability without compromising model accuracy.
> Further, references [15-16] conducted a comprehensive analysis of the attention mechanism, experimentally verified the overfitting phenomenon caused by the attention module. [15-16] strategically bypassed the attention mechanism to  not only maintain but can enhance the model performance.

---

> > ### Comment · Reviewer_9KoY · 2024-08-12
> >
> > Thanks to the authors' responses, our major concerns have been addressed.

---

> > > ### Author Response · Authors · 2024-08-13
> > >
> > > Dear Reviewer 9KoY:
> > >
> > > Thank you for your suggestions on experimental integrity and fairness, which help us conduct more rigorous research. We also appreciate your thoughtful feedback and the acknowledgment of the clarifications and new experiments. We sincerely value your careful consideration of our work.

---

### Author Rebuttal · Authors · 2024-08-06

1. In **Table 1** of our rebuttal pdf, three attack methods (including PNA, TGR and ours) are tested under different settings of input enhancement.
**“w / o”** denoted that no input enhancement was added, which is the same as the results shown in Tables 1-2 of our manuscript.
**“PO”** indicated that PatchOut was utilized as input enhancement.
**“SPPO”** indicated that Self-Paced Patch Out was utilized as input enhancement, with the same hyperparameters mentioned in our manuscript.
Due to the low ASR of PNA under the “SPPO” setting, we thought that our ATT method's hyperparameters were not compatible with PNA,
and thus we fine-tuned relevant hyperparameters and specified this input enhancement as **“SPPO+”**.
After this adjustment, the ASR of PNA (SPPO+) was significantly improved, which verified our observations.
This experiment validated the superiority of our ATT method compared to SOTAs in various attack settings.


2. Considering that the results in Tables 1-3 in our manuscript could reflect the effect of SPPO on ASR, we followed the suggestion of Reviewer 52P8's question 3.
In **Table 2** of our rebuttal pdf, we performed ablation experiments on three strategies (AVR/ST/HT), where **“AVR”** denoted Adaptive Variance Reduced Token Gradient; **“ST”** denoted soft truncation that we utilized the truncation factor to truncate Attention, QKV, and MLP modules; **“HT”** denoted hard truncation that we could truncate the gradient of selected Attention layers to zero.


3. In **Table 3** of our rebuttal pdf, we added a validation experiment for robust ViTs,
where PNA, TGR, and ATT (ours) use their optimal hyperparameters.
“clean” denoted that the model was utilized to classify a clean dataset and the result was the probability of the classifier classifying the data incorrectly.
The rest of the results were all ASRs.
Meanwhile, we performed the same test on the normal model corresponding to robust ViTs.
Experimental results showed that our ATT method still performs better than SOTAs in advanced normal and robust ViT models.


4. In **Table 4** of our rebuttal pdf, an additional experiment was conducted to study the variance reduction of token gradients, where both input enhancement and gradient strategies are not considered here.
**“Only VR”** denoted that only variance reduction strategy was utilized in our ATT method, while **“Only AVR”** denoted that only adaptive variance reduction (AVR) was ultized in our ATT method.
As shown in Table 4, our proposed adaptive variance reduction strategy performs better than the fixed variance reduction strategy, leading to better transferability of crafted adversarial examples.

---

### Decision · Program_Chairs · 2024-09-25

**Decision:**

Accept (poster)

**Comment:**

The paper proposes to enhance the transferability of adversarial attacks on Vision Transformers (ViTs) by introducing three main optimization strategies. The paper rececived two borderline accept, one borderline reject, and one weak accept. The main concerns are comparisons with existing approaches. The authors successfully addressed these concerns with a convincing rebuttal. The AC recommends acceptance and encourages the authors to revise the paper according the comments.